# Fast and Adaptive Multi-Objective Feature Selection for Classification

## Abstract

Identifying high-quality feature subsets for decision-makers by wrapper-based multi-objective feature selection (MOFS) has been attracting increasing attention. As a mainstream approach, evolutionary methods offer distinct advantages but also struggle with increasingly complex application scenarios and high-dimensional data, mainly in terms of time efficiency, exponentially expanding search space, and adaptive determination of a suitable classifier. To overcome these challenges, this paper proposes two simple yet effective methods: Fast Initialization (FI) and one-generation Adaptive K-Nearest Neighbor (AK). FI leverages mutual information and tournament selection to locate high-quality initial feature subsets computationally efficiently. AK verifies that, with theoretical proof, using a single generation can determine the most suitable KNN for different data to improve feature selection performance with very little time overhead and without any data analysis or assumption. Experiments on 20 real-world high-dimensional datasets demonstrate the superior performance of FI and AK to advanced initialization and KNN methods for MOFS. We also validated that the obtained feature subsets generalize well to an LLM for tabular data, enabling it to be seamlessly applied to high-dimensional data and achieve superior performance.

## 1 Introduction

In the era of increasing informatization and intelligence across various industries, classification tasks in real-world often encounter great challenges Wu et al. (2023) such as high-dimensional data Dong & Kluger (2023); Cheng et al. (2025), complex sample distributions (*e.g.,* imbalance and outliers), and the lack of interpretability in deep learning models. Feature selection (FS), as a data preprocessing technique, aims to identify small yet informative feature subsets. This not only enhances the interpretability of classification but also reduces training and inference costs, lowers data collection expenses, removes unimportant and noisy features, and mitigates overfitting to improve classification performance Xue et al. (2016).

In particular, multi-objective feature selection (MOFS) has attracted growing attention in recent years Jiao et al. (2024a), as it enables the discovery of a set of Pareto-optimal feature subsets, named Pareto front (PF), that represent trade-offs between classification performance and feature subset size. These trade-off solutions offer valuable flexibility for different decision-makers and application scenarios.

Wrapper-based FS using evolutionary computation (EC) methods has become a mainstream approach to MOFS due to the following reasons. First, real-world data often involve complex feature interactions. EC-based wrapper methods can effectively capture and retain these interactions during the search, leading to improved classification performance Van Der Maaten et al. (2009). Second, as a population-based search approach, EC is inherently well-suited for MOFS, offering both global search ability free from problem's analytical expression or gradient availability and the generation of diverse Pareto-optimal solutions in a single run Mukhopadhyay et al. (2014).

Despite the above advantages, it is still very challenging to handle high-dimensional data due to the following issues. First, evolutionary iterations and classification performance evaluation results in substantial time overhead Bian et al. (2020); Lu et al. (2024). Second, the limited number of evaluations greatly increases the difficulty of identifying high-quality subsets in high-dimensional feature spaces. Third, as a lazy model requiring no complex training process at each iteration and

is easy to implement with high performance, k-nearest neighbour (KNN) Cover & Hart (1967) is widely used as the classifier for wrapper-based feature selection in classification. However, real-world high-dimensional data exhibit complex and diverse characteristics, such as class imbalance and distributions of data points. Thus, using a fixed KNN with a deterministic $k$ value and ad-hoc neighbour voting mechanism is hard to handle complex data. Therefore, it is desired to address these challenges, improve the performance, and reduce the time consumption of evolutionary MOFS methods to better handle complex real-world high-dimensional data.

The initialization strategy plays a vital role in addressing the second issue since a proper initial population can facilitate the convergence in a high-dimensional search space and avoid getting trapped in local optima. However, existing initialization techniques in MOFS methods either fail to identify high-quality features through training data analysis, thus resulting in excessive randomness Han et al. (2023), or features are selected based solely on their importance with poor population diversity risking local optima Xu et al. (2021); Saadatmand & Akbarzadeh-T (2024), or the process is too time-consuming Jiao et al. (2024b); Cai & Xue (2024). For the third issue, considering the high computational cost of wrapper-based methods and the coherence between feature subset quality and the adopted classifier, it is desired yet challenging to automatically determine the most suitable KNN settings (k value and neighbor voting mechanism) for the given dataset, but without introducing much overhead. Although adaptive KNN selection or learning for classification has been well-studies Zhao & Lai (2021); Xie et al. (2024), existing methods are not tailored for wrapper-based MOFS and are thus inapplicable unless embedded into each iteration, which will inevitably cause unacceptable time consumption.

This paper addresses these issues and improves the effectiveness and efficiency of MOFS from both feature and classifier aspects[1]. The main contributions are as follows:

1). A **Fast Initialization (FI)** is proposed that measures the mutual information (MI) Shannon (1948); Gadgil et al. (2024) between each feature and class label as feature importance, and uses a parameter-adaptive tournament selection to balance feature importance and population diversity. FI achieves competitive accuracy compared to existing methods and is much more efficient.

2). An **Adaptive KNN (AK)** is developed to automatically determine the most suitable KNN for the given dataset with little overhead and without any assumption. Multiple independent populations evolve in the first generation, each using a KNN to preserve the coherence between features' quality and the classifier. Then, the population and its KNN settings with the best classification performance are selected to finish evolution.

3). The new method integrating FI and AK outperforms representative and advanced methods in experiments on 20 real-world high-dimensional datasets. Moreover, the obtained feature subsets generalize well to an LLM tailored for tabular data, which cannot be directly applied on these datasets due to high dimensionality. However, our feature selection method enables it to be directly applied to high-dimensional data and rapidly achieve superior performance.

## 2 RELATED WORK

**Initialization Methods in MOFS:** Existing initialization methods, based on whether the information of the data is used, can be divided into the following categories. A detailed review is presented in the Supplementary file.

Some methods propose to use the information extracted from the data to locate important features Wang et al. (2023b); Jiao et al. (2024b); Cai & Xue (2024); Hancer et al. (2024); Wang et al. (2023a). These methods use training data information to assist initialization, which might benefit high-dimensional MOFS. However, most of them introduce unacceptable overhead. Assuming the data contains $d$ features and $N$ samples, then the time consumption of calculating the correlation measure between each pair of features can be $O(d^2 N)$. Therefore, a time-efficient method is desired.

Quite a few evolutionary MOFS methods randomly generate the initial population Han et al. (2023); Song et al. (2024); Liang et al. (2024), which can significantly reduce time consumption but is ineffective in handling high-dimensional data. Others use population information to guide initialization Xu et al. (2021); Saadatmand & Akbarzadeh-T (2024); Cheng et al. (2024). In summary,

---

[1]In this work, the KNN selection aims to select a KNN for the whole dataset, rather than for each sample.

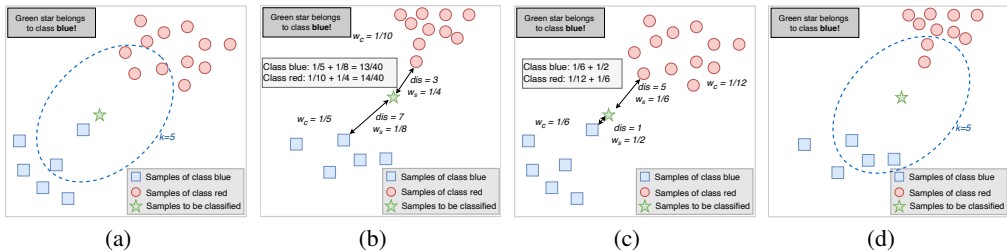

Figure 1: Limitations of using a fixed KNN on different data. (a). Using KNN with $k$ set to 5, where the classification result is incorrect; (b). Under the same situation as (a), KNN2W can correctly classify; (c). A situation where KNN2W obtains an incorrect classification result due to the incomplete pattern reflected by the insufficient samples of the minority class; (d). Under the same situation as (c), KNN can correctly classify. Such phenomena indicate that a fixed KNN in the MOFS algorithm is hard to handle distinct datasets.

random initialization has much lower time cost ($O(dN_p)$ where $N_p$ is the population size) and population information-based initialization can also be time-consuming (maximal $O(d^2 N_p)$ but usually after feature selection, $d$ is significantly reduced); however, due to the exponentially growing search space of high-dimensional problems, it is challenging to identify valuable features from a large number of features using limited individuals without utilizing the information from the training data.

**KNNs in MOFS for Classification:** Owing to the promising classification performance and easy-to-implement nature, KNN is widely used as the classifier in classification Xie et al. (2024). The common practice of using KNN in MOFS for classification includes two paths. One is using KNN with a fixed $k$ value Cheng et al. (2024); Jiao et al. (2024a), and another is improving the original KNN by a weighted approach such as the double-weighted KNN (named KNN2W) proposed in Saadatmand & Akbarzadeh-T (2024). However, as the dimensionality of the data increases and the distribution becomes complex, using a pre-defined and fixed KNN is less effective for different data. These limitations on complex data are illustrated by an artificial scene shown in Fig. 1 and analyzed in the Supplementary file.

In summary, the above situations show that when the feature dimensionality increases and the sample distribution becomes complex, using a single pre-defined KNN can be ineffective for different data.

## 3    PROPOSED METHODS

The MOFS problem in this work can be formulated as:

$$
\begin{aligned}
\text{Minimize} \quad & \mathbf{F}(\mathbf{x}) = (f_1(\mathbf{x}), f_2(\mathbf{x}))^T \\
\text{subject to} \quad & \mathbf{x} \in \mathbf{\Omega}^d
\end{aligned}
\tag{1}
$$

The balanced classification error $f_1$ is

$$
f_1(\mathbf{x}) = 1 - \frac{1}{c} \sum_{i=1}^{c} \frac{TP_i}{S_i},
\tag{2}
$$

where $c$ is the number of classes, $TP_i$ is the true positive samples in the $i$-th class, and $|S_i|$ is the number of samples in the $i$-th class. The weight for each class is set to $1/c$ to avoid biases to the majority classes for imbalanced data.

$f_2$ is the selected feature ratio calculated by

$$
f_2(\mathbf{x}) = \frac{\sum_{i=1}^{d} x_i}{d},
\tag{3}
$$

where $\mathbf{x} = (x_1, \ldots, x_d)^T$ is the decision vector with $d$ decision variables (*i.e.,* features); $x_i = 1$ means the $i$-th feature is selected, and $x_i = 0$ means not; $\mathbf{x} \in \mathbf{\Omega}^d$ is the search space (decision space), and $\mathbf{F} \in \mathbf{\Omega}^m$ is the objective space.

Figure 2: Overall flowchart of the **H**igh-dimensional **MOFS** framework based on our proposed **FI** and **AK** (HMOFS-FI-AK).

## 3.1 FRAMEWORK

The overall flowchart of our proposed high-dimensional MOFS framework based on FI and AK (HMOFS-FI-AK) is presented in Fig. 2. For a dataset with $N$ samples, each having $d$ features and one class label, we randomly split it into a training set (70%) and a test set (30%). Then, the MI is calculated based on the training set, and the proposed FI is conducted. Afterward, the initialized population is assigned to $t$ KNNs for evolution of one generation. Afterward, the KNN and population with the best minimal classification error is selected to finish the evolution. We then obtain the non-dominated solutions from the final population and evaluate them on the test set to select only the non-dominated feature subsets as the final output. So, HMOFS-FI-AK can obtain Pareto feature subsets for different practitioners and scenarios. For example, the LLM **TabPFN** Hollmann et al. (2025) tailored for tabular data strictly requires no more than 500 features.

## 3.2 FAST INITIALIZATION

---

**Algorithm 1** Fast Initialization

**Input:** $\mathcal{D}$ (Training Data), $N$ (# Samples), $d$ (# Features), $N_p$ (Population Size), $T$ (Sparse Factor)
**Output:** $\mathcal{P}$ (Initialized Population)
1: $MI \leftarrow$ Create an empty list with size $d$;
2: **for** $col = 1$ **to** $d$ **do**
3:     $MI(col) \leftarrow$ Calculate MI of $col$-th feature to class label by Equation equation 4;
4: **end for**
5: $\mathcal{P} \leftarrow$ Initialize the population as an $N_p \times d$ all-zero matrix;
6: **for** $i = 1$ **to** $N_p$ **do**
7:     Generate a random integer $k$, $k \in \{1, 2, \cdots, T\}$;
8:     $\mathbf{S} \leftarrow$ Initialize selected feature list as an empty list;
9:     **for** $j = 1$ **to** $\lceil T/k \rceil$ **do**
10:         $C \leftarrow$ Randomly choose $k$ distinct indices from $\{1, 2, \cdots, d\}$;
11:         $best \leftarrow \arg\max_{l \in C} MI(l)$;
12:         Append $best$ to $\mathbf{S}$;
13:     **end for**
14:     $\mathcal{P}(i, \mathbf{S}) \leftarrow 1$;
15: **end for**
16: $\mathcal{P} \leftarrow$ Delete redundant feature subsets;
17: **return** $\mathcal{P}$.

---

The pseudocode of FI is summarized in Algorithm 1. The MI between the $j$-th feature $X_j$ and the class label $Y$ is calculated by:

$$I(X_j; Y) = \sum_{x \in \mathcal{X}_j} \sum_{y \in \mathcal{Y}} \frac{n_{x,y}}{N} \log\left(\frac{n_{x,y} \cdot N}{n_x \cdot n_y}\right), \quad (4)$$

where $X_j$ is the $j$-th feature column, for $j = 1, 2, \ldots, d$; $Y$ is the class label column; $\mathcal{X}_j$ is the set of unique values taken by feature $X_j$; $\mathcal{Y}$ is the set of all possible class labels; $n_{x,y}$ is the number of samples where $X_j = x$ and $Y = y$; $n_x$ is the number of samples where $X_j = x$; $n_y$ is the number of samples where $Y = y$; $N$ is the total number of samples.

The probabilities $p(x)$, $p(y)$, and $p(x,y)$ are estimated empirically from the dataset using frequency counts, *i.e.*,

$$p(x,y) \approx \frac{n_{x,y}}{N}, \quad p(x) \approx \frac{n_x}{N}, \quad p(y) \approx \frac{n_y}{N}. \quad (5)$$

For continuous features, we simply discretize them using the equal-frequency with 10 bins.

Since we don't calculate the MI between features, the time consumption is much less than existing MI-based methods. Moreover, a parameter-adaptive mechanism is designed (lines 7-14), which leverages the MI and tournament selection to balance feature importance and population diversity. Specifically, a large $k$ leads to a small feature subset ($\lceil T/k \rceil$). Therefore, we select more important

---

**Algorithm 2** MOFS Framework based on Adaptive KNN

---

**Input:** $MOFS$ (Adopted MOFS Algorithm for Evolution), $\{KNN_1, \cdots, KNN_t\}$ (KNNs)
**Output:** $\mathcal{P}$ (Final Population, *i.e.,* Feature Subsets)
1: $\mathcal{P} \leftarrow$ Initialize a population using FI by Algorithm 1;
2: Assign $\mathcal{P}$ to populations $\mathcal{P}_1$ to $\mathcal{P}_t$ and correspondingly use each KNN: $KNN_1$ to $KNN_t$ as classifier to evaluate;
3: $\mathcal{P}_1$ to $\mathcal{P}_t$ evolve one generation independently by $MOFS$ algorithm and the corresponding KNN;
4: $\mathcal{P}, KNN \leftarrow$ Determine the population with the lowest minimal classification error among all individuals, and get its KNN;
5: $\mathcal{P} \leftarrow$ Finish the evolution starting with $\mathcal{P}$ and using $KNN$ by $MOFS$;
6: $\mathcal{P} \leftarrow$ Evaluate on test data and select non-dominated solutions as the final PF;
7: **return** $\mathcal{P}$ as PF.

---

features by a $k$-tournament that selects the feature with the largest MI among $k$ randomly selected features. On the contrary, a small $k$ leads to a large feature subset. Then, we select more diverse features by $k$-tournament from a few randomly selected features. Consequently, small feature subsets contain important features to improve classification performance, while large feature subsets provide diversity for exploring the search space.

### 3.3 ADAPTIVE KNN

The pseudocode of the proposed AK is summarized in Algorithm 2. While wrapper-based EC methods are effective, their high computational cost renders the use of classifier ensembles often infeasible in practical scenarios. However, in AK, all populations only evolve one generation using their corresponding KNNs. Afterward, the most suitable KNN is determined according to the minimal $f_1$ value (classification error). Therefore, this simple mechanism introduces only little overhead, requires no data analysis or assumption, and is easy to extend or implement.

The underlying principles and theoretical analysis of AK are as follows. Detailed proof is presented in the Supplementary file.

Let $\mathcal{P}^{(t)}$ denote the population of feature subsets in generation $t$; meanwhile, let $p = 1/d$ be the per-bit mutation rate on a $d$-dimensional feature string. We model the population sequence $\{\mathcal{P}^{(t)}\}$ as a finite–state Markov chain Norris (1998). The mutation step flips a binomial number of bits whose tail probability is exponentially small by the Chernoff bound Chernoff (1952); Hoeffding (1963), so offspring remain within an $O(\ln d)$ Hamming ball of their parents with high probability. Moreover, the expected progress of the best individual can be bounded using the locality of evolutionary operators and drift analysis He & Yao (2001); Shastri & Frachtenberg (2020), which characterizes the expected one–step decrease of a distance-to-optimum potential function and implies that the search process concentrates around the local optimum at a rate governed by the mutation strength. With classifier accuracy assumed $L$–Lipschitz in Hamming distance and the initial population covering a $\delta$-radius neighborhood of the local optimum with fraction $\rho$ (i.e., at least a $\rho$ proportion of individuals lie within Hamming distance $\delta$ of the local maximizer), the empirical accuracy gap $\Delta$ between the best configuration and all others cannot shrink by more than $O(\delta)$ per generation. If the initial margin satisfies $\rho\Delta > 4\rho L\delta$, a union bound over $G$ generations (each generation violating the locality or drift condition with probability at most $\eta$, where $\eta$ bounds the chance that mutation or crossover escapes the $\delta$-ball) guarantees that the configuration achieving the largest initial accuracy remains the unique maximizer with probability at least $1-(G\eta+\xi)$. This establishes one–generation identifiability and multi–generation stability of the selected KNN configuration.

## 4 EXPERIMENTAL SETTINGS

In this paper, we embed FI and AK into the DAEA Xu et al. (2021) algorithm. Then, we compare FI with random initialization and the initialization methods of DAEA Xu et al. (2021), SMII Cai & Xue (2024), JSEMO Saadatmand & Akbarzadeh-T (2024), and FPPFS Jiao et al. (2024b), and compare AK to KNN ($k = 5$) and KNN2W (denoted as JSCla). Experiments were conducted on PlatEMO Tian et al. (2017) and TabPFN Hollmann et al. (2025) using a local computer with Intel Ultra9 285K CPU, 32 GB RAM, and RTX 5080 16G in Windows 11 operating system.

**Datasets:** We adopt 20 real-world high-dimensional datasets whose detailed information is presented in the Supplementary file. The datasets are all open-source data. Due to the large dimensions of most data sets, the experiments were extremely time-consuming for Cai & Xue (2024); Jiao et al. (2024b), so only several datasets are used in these two algorithms.

**Parameter Settings of the EA:** Following most MOFS studies, the evolutionary parameters are set as follows. The population size $N_p$ is 200. The maximal number of individual evaluations $E_{max}$ is 20000, and the maximal generations $G_{max}$ is $\lceil E_{max}/N_p \rceil$. In our methods, FI contains a parameter $T$, which is set to 400. All parameters of other algorithms and methods are the same as in their original literature.

**Performance Indicators:** The hypervolume (HV) Zitzler & Thiele (1998) is adopted to measure the performance in the multi-objective optimization aspect. Since the true PFs are unknown, we save the final results of all algorithms. Then, we extract the non-dominated solutions and use their objective vectors to get the ideal and nadir points for normalization. After normalization, we use (1.1,...,1.1) as the reference point for HV calculation. In addition, we also record the minimum classification error (MCE) of the final population of each algorithm as the classification performance.

**Statistical Analysis:** Each algorithm independently executes 30 times on each dataset. The mean and standard deviation values of HV and MCE are recorded. The Wilcoxon rank-sum test with a significance level of 0.05 is employed. "+", "−", and "≈" indicate the result of another algorithm is significantly better than, significantly worse than, and statistically similar to our methods. The Friedman test with a Holm correction at a significance level of 0.05 is also used to obtain rankings and p-values of different algorithms.

## 5 RESULTS AND ANALYSES

### 5.1 FI AND EXISTING INITIALIZATION METHODS

The results of FI compared to existing methods in MOFS literature and random initialization are summarized in Table 1. FI achieves better HV and MCE results than all other methods, showcasing a significant advantage in HV. The HV advantage means HMOFS-FI finally obtains a Pareto feature subset with significantly better convergence (minimization of both classification error and subset size) and diversity (more Pareto solutions). This indicates that FI can provide a high-quality initial population to facilitate the convergence and diversity of evolutionary search. Moreover, on CPU run time, FI also achieves promising results. Although DAEA performs better on run time, it performs significantly worse on HV and slightly worse on MCE.

| | HV | | MCE | | Time | |
|---|---|---|---|---|---|---|
| | Ranking | +/-/= | Ranking | +/-/= | Ranking | +/-/= |
| HMOFS-DAEA | 2.225 | 0/12/8 | 2.35 | 0/1/19 | **1.05** | 19/1/0 |
| HMOFS-JSEMO | 2.5 | 0/10/10 | 2.175 | 0/4/16 | 3.15 | 1/19/0 |
| HMOFS-Random | 3.575 | 0/17/3 | 3.5 | 0/13/7 | 3.6 | 4/16/0 |
| HMOFS-FI | **1.8** | | **1.925** | | 2.2 | |

Table 1: Results of FI and other methods, including the Friedman rankings and Wilcoxon test results.

Fig. 3 depicts the mean values of CPU run time of each method on each dataset. From the results, FI is much more time-efficient than other methods, especially on datasets with very high dimensionality. Among the peer initialization methods, JSEMO can be time-consuming, while DAEA is time-efficient. Nevertheless, the initialization methods of JSEMO and DAEA both perform worse.

In this part, we also conduct comparisons including SMII and FPPFS on several relatively low-dimensional datasets. Detailed results and analysis are presented in the Supplementary file. It can be observed that

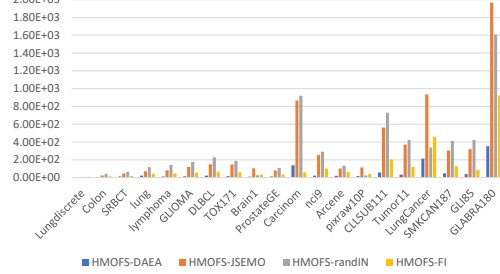

Figure 3: CPU run time of using different initialization methods on all datasets.

the time overhead of SMII and FPPFS increases sharply with the dimensionality, reaching an unacceptable level. In contrast, DAEA and FI are less sensitive to dimensionality changes. However,

the initialization strategy of DAEA does not consider information from the data, resulting in inferior performance compared to FI in terms of HV and MCE.

In summary, FI can provide a high-quality initial population for evolutionary MOFS search in a time-efficient way. Detailed results are presented in the Supplementary file.

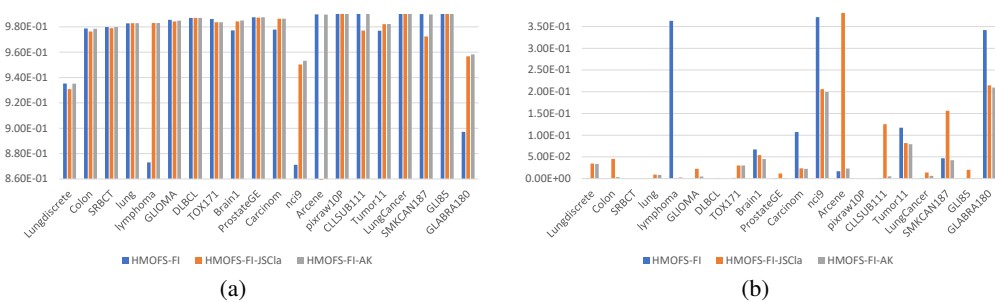

(a)                                                                                                              (b)

Figure 4: HV (a) and MCE (b) of using KNN ($k = 5$), KNN2W, and our proposed AK method.

## 5.2 AK AND SINGLE FIXED KNN

The HV and MCE results of using KNN ($k = 5$) or KNN2W and AK to adaptively select from them are presented in Fig. 4. Although HMOFS-FI-AK does not achieve the best performance on every dataset, the HV comparisons on each dataset show that HMOFS-FI-AK consistently outperforms the worse one between KNN ($k = 5$) and KNN2W, indicating that AK is capable of selecting the more suitable KNN. The slight performance degeneration is attributed to the fact that different KNNs consume part of the very limited evaluation budget in the first generation.

The difference is more pronounced in terms of MCE: when the more suitable KNN is selected for a given dataset, the final MCE can be significantly reduced. Similarly, AK consistently achieves better results than the worst-performing KNN, indicating its ability to select the more appropriate KNN. The slightly inferior results are due to the same reason mentioned above.

In summary, AK can adaptively determine the most suitable KNN based on the experimental results. Detailed results are presented in the Supplementary file.

## 5.3 COMPARISONS ON PFs AND CONVERGENCE PROFILES

To intuitively compare the final PFs, we select eight representative datasets and plot the final PFs using different initialization methods and KNNs in Fig. 5.

SRBCT is of relatively low dimensionality, so the performance of HMOFS-FI-AK is just slightly better than HMOFS-FI. This reveals that FI performs better and KNN ($k = 5$) is more suitable. Moreover, it also verifies that AK selects the correct KNN, so it performs much better than HMOFS-FI-JSCla. On lymphoma, it can be found that KNN2W is much more effective, AK can select KNN2W, and FI can help improve the convergence. On TOX171, HMOFS-FI and HMOFS-JSEMO perform well, and HMOFS-FI-AK can determine KNN ($k = 5$). On ProstateGE, HMOFS-FI and HMOFS-FI-AK achieve the best performance, revealing that FI is effective and KNN ($k = 5$) is determined by AK. On Carcinom and nci9, HMOFS-FI-AK obtains the best performance with KNN2W selected. On Tumor11, using KNN ($k = 5$) achieves better overall convergence but KNN2W brings better classification performance. AK somehow achieves a trade-off between them. On GLABRA180, HMOFS-FI-AK obtains the best PF, where FI performs similar to DAEA, but AK significantly improves the final performance. [2]

The convergence profiles of HV and MCE on five representative datasets are plotted in Fig. 6. On Brain1, nci9, Tumor11, and GLABRA180, KNN2W is more suitable, and AK can determine this at the first generation and then quickly catch up with its performance and eventually achieve compara-

---

[2]Comparisons on convergence profiles are presented in the Supplementary file.

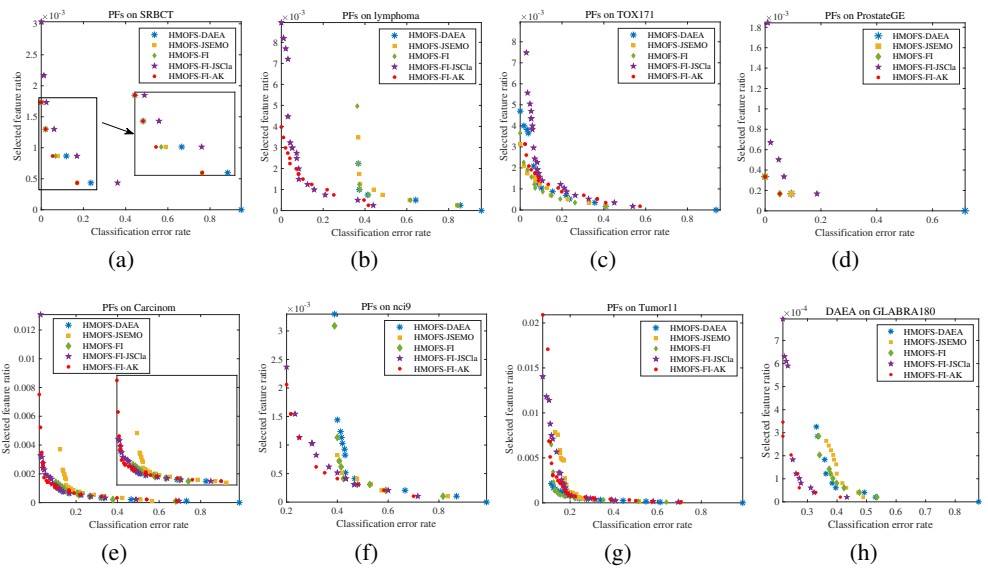

Figure 5: PFs obtained by different methods on test sets of representative datasets with the median HV values among the 30 runs.

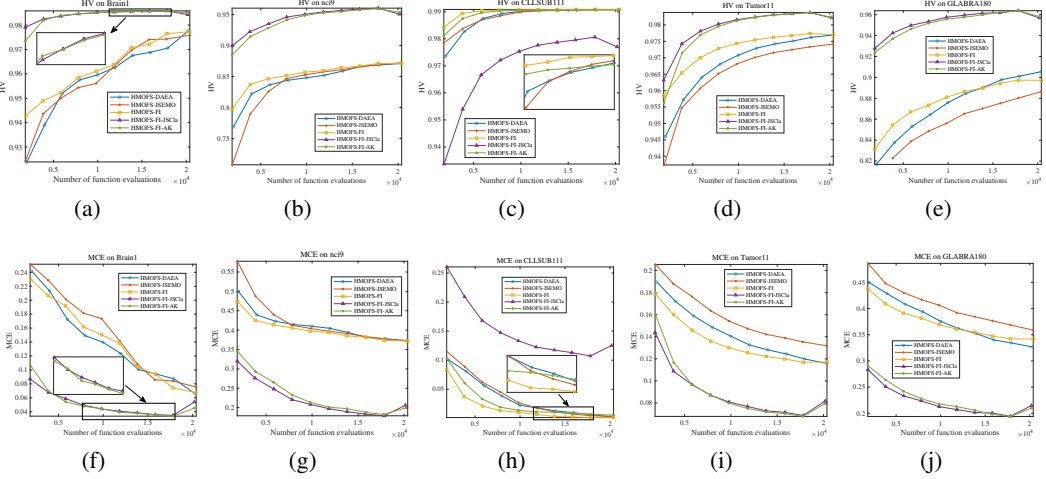

Figure 6: Convergence profiles of HV and MCE indicators of different methods on five representative datasets with the median indicator values among the 30 runs. The final step shows the performance changes from the training to the test set.

ble or even better results. On CLLSUB111, KNN ($k = 5$) is more suitable and AK can also detect. On all datasets, the header start and better convergence performance verify the effectiveness of FI.

## 5.4 APPLYING TO LLM TABPFN

To verify that our MOFS results can generalize well to popular deep learning models, we obtain the final feature subset of each algorithm with the best balanced accuracy, including the original DAEA and JSEMO algorithms. Then, we use these features to form data subsets. Afterward, these data subsets can be directly used in LLM **TabPFN**. It should be noted that LLM **TabPFN** cannot be directly used on these datasets due to the high dimensionality. TabPFN provides different measures: ROC AUC, balanced accuracy, and accuracy. The results are presented in Table 2. On most datasets, FI-AK achieves the best results as FI provides a better initial population and AK determines a suitable KNN; on Carcinom and SMK-CAN-187, FI or FI-JSCla obtains the best, while FI-AK ranks

Table 2 (left half):

| Dataset | Measure | JSEMO | DAEA | FI | FI-JSCla | FI-AK |
|---|---|---|---|---|---|---|
| lung_discrete | ROC AUC | **0.99** | **0.99** | 0.97 | **0.99** | **0.99** |
| | Balanced Acc | 0.94 | 0.89 | 0.67 | **0.96** | **0.96** |
| | Accuracy | 0.92 | 0.92 | 0.72 | **0.96** | **0.96** |
| Colon | Balanced Acc | 0.89 | 0.85 | **0.96** | 0.82 | 0.89 |
| | Accuracy | 0.90 | 0.85 | **0.95** | 0.85 | 0.90 |
| SRBCT | ROC AUC | **1.00** | **1.00** | 0.99 | **1.00** | **1.00** |
| | Balanced Acc | 0.95 | 0.95 | 0.85 | **1.00** | **1.00** |
| | Accuracy | 0.96 | 0.96 | 0.86 | **1.00** | **1.00** |
| lung | ROC AUC | 0.98 | **0.99** | **0.99** | **0.99** | **0.99** |
| | Balanced Acc | 0.89 | 0.96 | 0.89 | **0.99** | 0.96 |
| | Accuracy | 0.89 | 0.95 | 0.92 | **0.97** | 0.96 |
| lymphoma | ROC AUC | 0.95 | 0.88 | 0.92 | **0.99** | 0.97 |
| | Balanced Acc | 0.81 | 0.78 | 0.77 | 0.81 | **0.85** |
| | Accuracy | 0.90 | 0.87 | 0.87 | 0.90 | **0.94** |
| GLIOMA | ROC AUC | 0.91 | 0.91 | 0.84 | 0.96 | **0.97** |
| | Balanced Acc | 0.65 | 0.62 | 0.50 | **0.95** | 0.82 |
| | Accuracy | 0.76 | 0.64 | 0.58 | **0.94** | 0.88 |
| DLBCL | Balanced Acc | 0.69 | 0.75 | **1.00** | 0.97 | **1.00** |
| | Accuracy | 0.70 | 0.75 | **1.00** | 0.96 | **1.00** |
| TOX-171 | ROC AUC | 0.95 | 0.94 | 0.98 | 0.95 | **0.99** |
| | Balanced Acc | 0.77 | 0.82 | 0.82 | 0.75 | **0.88** |
| | Accuracy | 0.77 | 0.82 | 0.82 | 0.75 | **0.88** |
| Brain1 | ROC AUC | 0.96 | 0.85 | 0.95 | **0.97** | 0.95 |
| | Balanced Acc | 0.80 | 0.73 | 0.53 | 0.88 | **0.89** |
| | Accuracy | **0.93** | 0.90 | 0.80 | 0.90 | **0.93** |
| Prostate-GE | Balanced Acc | 0.88 | 0.94 | **0.97** | **0.97** | **0.97** |
| | Accuracy | 0.88 | 0.94 | **0.97** | **0.97** | **0.97** |

Table 2 (right half):

| Dataset | Measure | JSEMO | DAEA | FI | FI-JSCla | FI-AK |
|---|---|---|---|---|---|---|
| Carcinom | ROC AUC | 0.99 | 0.96 | 0.99 | **1.00** | 0.99 |
| | Balanced Acc | 0.86 | 0.83 | 0.83 | **0.97** | 0.87 |
| | Accuracy | 0.91 | 0.87 | 0.89 | **0.96** | 0.93 |
| nci9 | Balanced Acc | 0.50 | 0.27 | 0.54 | **0.63** | **0.63** |
| | Accuracy | 0.50 | 0.25 | 0.55 | **0.60** | **0.60** |
| arcene | Balanced Acc | 0.56 | 0.89 | 0.89 | 0.50 | **0.93** |
| | Accuracy | 0.56 | 0.89 | 0.89 | 0.56 | **0.93** |
| pixraw10P | ROC AUC | **1.00** | **1.00** | **1.00** | **1.00** | **1.00** |
| | Balanced Acc | 0.97 | **1.00** | **1.00** | 0.97 | **1.00** |
| | Accuracy | 0.96 | **1.00** | **1.00** | 0.96 | **1.00** |
| CLL_SUB_111 | ROC AUC | 0.89 | 0.86 | 0.87 | 0.83 | **0.90** |
| | Balanced Acc | 0.83 | 0.83 | 0.73 | 0.81 | **0.86** |
| | Accuracy | 0.79 | 0.78 | 0.64 | 0.75 | **0.81** |
| Tumor11 | ROC AUC | 0.98 | 0.95 | **0.99** | **0.99** | **0.99** |
| | Balanced Acc | 0.84 | 0.75 | 0.83 | 0.77 | **0.87** |
| | Accuracy | 0.86 | 0.79 | 0.87 | 0.82 | **0.89** |
| Lung_Cancer | ROC AUC | **0.99** | 0.98 | 0.95 | **0.99** | **0.99** |
| | Balanced Acc | **0.94** | 0.87 | 0.90 | 0.89 | **0.94** |
| | Accuracy | **0.97** | 0.94 | 0.95 | 0.94 | **0.97** |
| SMK-CAN-187 | Balanced Acc | 0.73 | 0.70 | **0.80** | 0.72 | 0.77 |
| | Accuracy | 0.74 | 0.70 | **0.80** | 0.72 | 0.77 |
| GLI_85 | Balanced Acc | 0.86 | 0.80 | 0.81 | 0.86 | **0.94** |
| | Accuracy | 0.89 | 0.86 | 0.82 | 0.89 | **0.97** |
| GLA_BRA_180 | ROC AUC | 0.88 | 0.86 | 0.86 | **0.89** | **0.89** |
| | Balanced Acc | 0.65 | 0.64 | 0.64 | 0.64 | **0.79** |
| | Accuracy | 0.73 | 0.70 | 0.70 | 0.71 | **0.81** |

Table 2: **ROC AUC, Balanced Accuracy, and Accuracy** results of important peer methods and our proposed methods using their obtained feature subsets for TabPFN.

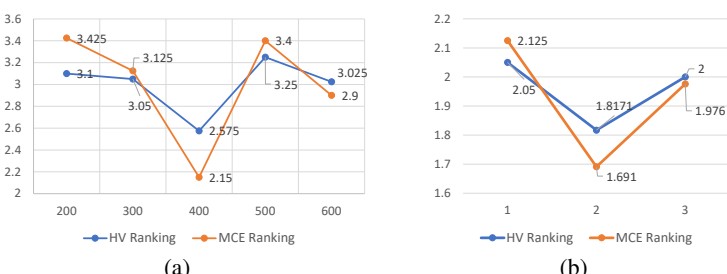

(a)  (b)

Figure 7: Friedman test summary of: (a). parameter studies on $T$ in FI and (b). the number of generations for all KNNs used in AK.

second. These results demonstrate that FI is effective and AK can determine the most suitable KNN to assist in searching for high-quality feature subsets that generalize well to other models. It is also worth noting that after feature selection, TabPFN can finish its inference in a few seconds on all these datasets. Therefore, it also demonstrates that feature selection is promising in significantly reducing the time-consumption of LLMs in fine-tuning and inference.

## 5.5 Sensitivity Analysis

In FI, $T$ is used to control the sparsity, so its sensitivity is analyzed. In AK, we also conduct parameter studies using zero or two generations. Friedman test is performed on the HV and MCE results and summarized in Fig. 7. $T = 400$ in FI performs significantly better, and one generation in AK performs significantly better. Detailed results are presented in the Supplementary file.

## 6 Conclusions

This paper proposes two techniques, FI and AK, for evolutionary MOFS for high-dimensional data classification. Based on experiments on 20 real-world datasets, our proposed HMOFS-FI-AK framework can obtain Pareto feature subsets for different scenarios in a time-efficient way. The two techniques are time efficient, simple yet effective, and easy to implement and extend. Experiments on an LLM **TabPFN** for tabular data verify that HMOFS-FI-AK can obtain effective feature subsets, enabling it to be seamlessly applied to high-dimensional data and achieve superior accuracy and time efficiency.

In the future, it is promising to apply our methods to high-dimensional data in different areas Blumberg et al. (2024); Ong et al. (2025) to facilitate the use of LLMs and other advanced techniques.

## ETHICS STATEMENT

This study primarily focuses on improving machine learning models and algorithmic methodologies, without involving human subjects, personally identifiable data, or potential manipulation/monitoring scenarios. We consider that this research poses no significant ethical risks and have therefore not designed specific ethical intervention mechanisms. If this technology is applied to concrete applications in the future (such as automated decision-making systems or recommendation systems), developers should carefully consider appropriate privacy, security, and fairness constraints.

## REPRODUCIBILITY STATEMENT

We provide key details of the datasets used in this study in the supplementary material, and disclose the training/testing split, experimental settings, and pseudocode in the main text. We also anonymously release the HMOFS-FIAK code and the data after feature selection used for TabPFN on an anonymous GitHub link here, enabling direct reproduction of the experiments. The code and data are submitted to ICLR2026 as well.

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

# This is the Supplementary file of "Fast and Adaptive Multi-Objective Feature Selection for Classification".

## A INITIALIZATION METHODS IN MOFS

### A.1 TRAINING DATA INFORMATION-BASED INITIALIZATION

Some methods propose to use the information extracted from the training data to locate important features. Wang *et al.* Wang et al. (2023b) developed a heuristic information for ant colony optimization for MOFS by integrating symmetric uncertainty between each feature and class label and MI between each feature pair, then used this heuristic information in initializing an ant colony. Calculating the symmetric uncertainty between features and class labels is manageable, but the computation cost of MI between every pair of features is very high. Jiao *et al.* Jiao et al. (2024b) suggested to use multiple filter-based measures to construct performance predictor for preselecting feature subsets. Only the promising feature subsets preselected go into function evaluation. However, the predictor uses multiple filter measures including the calculation of information between feature and class label, as well as between features. Thereby, although function evaluation is saved, the overhead of preselection in initialization and other parts is very high. Cai *et al.* Cai & Xue (2024) proposed an initialization method based on Jaccard similarity between individuals to improve the diversity of the initial population and MI between features to locate important features, and the MI calculation is extremely time-consuming. Hancer *et al.* Hancer et al. (2024) devised a correlation-based feature elimination strategy that eliminates features with low correlation to the class label and high correlation with other features. Then, initialization is conducted in the reduced feature space. Since the correlation between each feature pair needs to be calculated, this method could be time-consuming. Wang *et al.* Wang et al. (2023a) sorted the features using the maximal information coefficient (MIC) between each feature and the class label that evaluates the importance of each feature. Subsequently, in the first half of the population, the $i$-th individual selects the first $i$ features; in each individual of the second half of the population, each feature is considered. A random number is generated, and if it is greater than the MIC value of that feature, the feature is selected. However, the calculation of MIC requires complex grid separation and calculation of information entropy, which are also time-consuming.

These methods use training data information to assist intialization, which might benefit high-dimensional MOFS. However, most of them introduce unacceptable overhead. Assuming the data contains $d$ features and $N$ samples, then the time consumption of calculating correlation measure between each pair of features can be $O(d^2N)$.

### A.2 RANDOM-BASED OR POPULATION INFORMATION-BASED INITIALIZATION

Quite a few evolutionary MOFS methods randomly generate initial population Han et al. (2023); Song et al. (2024); Liang et al. (2024), which can highly reduce time consumption but is ineffective in handling high-dimensional data. Early in 2021, Xu *et al.* Xu et al. (2021) improved the random initialization method by adding sparsity control by sampling only a small group from the entire feature set. In this way, the sparsity of feature subsets is enhanced but it lacks utilization of training data information. Very recently, they proposed another method Xu et al. (2024) that simultaneously generate multiple random populations to cover the objective space, and then select a well-distributed set of non-dominated solutions from them as the initial population. This method also does not utilize the information from the training data, and multiple populations will consume a significant number of function evaluations, which are important in high-dimensional problems. Saadatmand *et al.* Saadatmand & Akbarzadeh-T (2024) used the Jaccard similarity in initialization by reducing the duplication with the already generated individuals using Jaccard similarity when generating each initial individual, Similarly, the training data information is not utilized and the performance might be degenerated in high-dimensional search space. This method is also time-consuming due to the calculation of Jaccard similarity in high-dimensional space. Cheng *et al.* Cheng et al. (2024) presented a reinitialization method that reinitializes the population when the minimum error obtained by the population does not change over multiple iterations to escape local optima. However, randomly restarting the entire population frequently may lead to unstable algorithm performance and waste function evaluations, which is particularly detrimental for solving high-dimensional problems.

| Index | Dataset | NO. of Fea | NO. of Instance | NO. of Class | Imbalance Rate |
|-------|---------|------------|-----------------|--------------|----------------|
| 1 | lung_discrete | 325 | 73 | 7 | 4.2 |
| 2 | Colon | 2000 | 62 | 2 | 1.82 |
| 3 | SRBCT | 2308 | 83 | 4 | 2.64 |
| 4 | lung | 3312 | 203 | 5 | 23.17 |
| 5 | lymphoma | 4026 | 96 | 9 | 23 |
| 6 | GLIOMA | 4434 | 50 | 4 | 2.14 |
| 7 | DLBCL | 5469 | 77 | 2 | 3.05 |
| 8 | TOX-171 | 5748 | 171 | 4 | 1.15 |
| 9 | Brain1 | 5920 | 90 | 5 | 15 |
| 10 | Prostate-GE | 5966 | 102 | 2 | 1.04 |
| 11 | Carcinom | 9182 | 174 | 11 | 4.5 |
| 12 | nci9 | 9712 | 60 | 9 | 4.5 |
| 13 | arcene | 10000 | 200 | 2 | 1.27 |
| 14 | pixraw10P | 10000 | 100 | 10 | 1 |
| 15 | CLL_SUB_111 | 11340 | 111 | 3 | 4.64 |
| 16 | 11Tumor | 12533 | 174 | 11 | 4.33 |
| 17 | Lung_Cancer | 12600 | 203 | 5 | 23.17 |
| 18 | SMK-CAN-187 | 19993 | 187 | 2 | 1.08 |
| 19 | GLI_85 | 22283 | 85 | 2 | 2.27 |
| 20 | GLA_BRA_180 | 49151 | 180 | 4 | 3.52 |

Table 3: Detailed information of the adopted datasets.

In summary, random initialization has much lower time cost ($O(dN_P)$) and population information-based initialization can also be time-consuming (maximal $O(d^2N_P)$ but usually after feature selection, $d$ is significantly reduced); however, due to the exponentially growing search space of high-dimensional problems, it is challenging to identify valuable features from a large number of features using limited population individuals without utilizing the information from the training data.

## B    KNNs in MOFS for Classification

The limitations of using fixed KNN on complex data are illustrated by an artificial scene shown in Fig. 1, and detailed illustrations are supplemented below. This situation includes two classes marked as class blue with fewer samples and class red with more samples, and the sample to be classified is marked as green star, which belongs to the class blue. In (a), the sample density of the blue class is lower than that of the red class. In (b), since the blue class is a minority class, it only reflects part of the pattern, resulting in its distance to class red smaller than that to class blue. From Fig. 1.(a), a fixed $k$ value can result in incorrect classification when the distribution of data is complex. From Fig. 1.(b), the KNN with two weights (KNN2W) proposed for imbalanced data classification in Saadatmand & Akbarzadeh-T (2024), where $k$ is fixed to one, can also result in incorrect classification due to the uneven distribution of samples. In this circumstance, using KNN with $k = 5$ may be better even for such imbalanced data.

## C    Datasets for Experiments

**Datasets:** We adopt 20 real-world high-dimensional datasets whose detailed information is presented in the Supplementary file. The data sets are all open source data downloaded from http://archive.ics.uci.edu and https://jundongl.github.io/scikitfeature/ data sets.html. Due to the large dimensions of most data sets, the experiments were extremely time-consuming for Cai & Xue (2024); Jiao et al. (2024b), so only several datasets are used in these two algorithms. We randomly split each data set into training a data set and a test data set with proportions of 70% and 30%, respectively.

## D    CPU Run Time Comparisons of Six Initialization Methods on Five Datasets

In the above experiments, SMII and FPPFS are not conducted on all datasets because they are extremely time-consuming on high-dimensional datasets, which hinders their feasibility in many real-world applications. In this part, we conduct comparisons including SMII and FPPFS on sev-

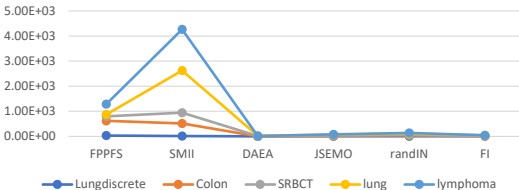

Figure 8: CPU run time of using different initialization methods on the first five datasets.

eral relatively low-dimensional datasets. The CPU run time results are presented in Fig. 8. It can be observed that the time overhead of SMII and FPPFS increases sharply with the dimensionality, reaching an unacceptable level. In contrast, DAEA and FI are less sensitive to dimensionality changes. However, the initialization strategy of DAEA does not consider information from the data, resulting in inferior performance compared to FI in terms of HV and MCE.

# Following is the theoretical proof of AK.

It should be noted that, to improve the readability of the argument, we use notations in the following proof that differ from those used in the proposed techniques.

## A PROBLEM STATEMENT

In wrapper-based evolutionary feature selection, K-Nearest Neighbors (KNN) is widely used as classifier. The KNN configurations significantly affect the evaluation results of feature subsets. We focus on different configurations of the KNN classifier, which is highly sensitive to feature selection because distance-based classification deteriorates in high dimensions Beyer et al. (1999).

Given a shared initial population of feature subsets for several KNN classifiers, we aim to demonstrate that:

> *It is theoretically justifiable to identify the most search-compatible KNN config-*
> *uration for a given dataset after only one generation of evolutionary evaluation,*
> *when the feature space is high-dimensional and the search budget is limited.*

## B SETTINGS AND NOTATIONS

Let $F = \{f_1, \ldots, f_n\}$ be the full feature set ($n$ large). Let $\mathcal{K} = \{K_1, \ldots, K_m\}$ be a finite set of KNN configurations. For generation $t$ denote the population by

$$P^{(t)} = \{S_1^{(t)}, \ldots, S_T^{(t)}\}, \qquad |P^{(t)}| = T.$$

For a configuration $K_j$ and subset $S$ write $\mathrm{Acc}_{K_j}(S) \in [0, 1]$ for validation accuracy and

$$M_j^{(t)} = \max_{S \in P^{(t)}} \mathrm{Acc}_{K_j}(S).$$

Hamming distance is $d_H(S, S')$. Throughout we treat $P^{(0)}$ as the shared initial population used to compare configurations.

## C ASSUMPTIONS

**Assumption 1** (Configuration diversity). *For any $i \neq j$ there exists some $S$ with $\mathrm{Acc}_{K_i}(S) \neq \mathrm{Acc}_{K_j}(S)$.*

**Assumption 2** (Local Lipschitz sensitivity (probabilistic)). *There exists a constant $L > 0$ and a failure rate $\xi \in (0, 1)$ such that for every configuration $K_j$ and all $S, S' \in \{0, 1\}^n$,*

$$\Pr\left[\left|\mathrm{Acc}_{K_j}(S) - \mathrm{Acc}_{K_j}(S')\right| \leq L\, d_H(S, S')\right] \geq 1 - \xi.$$

*This probabilistic form is more realistic for high-dimensional KNN.*

**Assumption 3** (Operator locality). *There exists $\delta \geq 1$ and $\eta \in (0, 1)$ such that for any parent $S$ the offspring $S'$ produced by mutation (with typical rate $p$) and crossover satisfies*

$$\Pr\left(d_H(S, S') \leq \delta\right) \geq 1 - \eta,$$

*a property well-studied in evolutionary computation Bäck et al. (1997); He & Yao (2001); Rothlauf (2002); Shastri & Frachtenberg (2020); M"uhlenbein (1992).*

**Assumption 4** (Local coverage). *For each configuration $K_j$ let $S_j^* \in \arg\max_{S \in P^{(0)}} \mathrm{Acc}_{K_j}(S)$. Assume there exists $\rho \in (0, 1]$ such that the Hamming ball $B(S_j^*, \delta)$ contains at least $\rho T$ members of $P^{(0)}$.*

## D CONCENTRATION: CHERNOFF AND HOEFFDING BOUNDS

To bound mutation step sizes, we use the classical Chernoff bound Chernoff (1952); Hoeffding (1963). Consider $D \sim \mathrm{Binomial}(n, p)$ with mean $\mu = np$.

**Proposition 1** (Chernoff bound—mgf derivation). *For any $\delta > 0$,*

$$\Pr\left(D \geq (1+\delta)\mu\right) \leq \exp\left(-\mu\left[(1+\delta)\ln(1+\delta) - \delta\right]\right).$$

*Consequently,*

- *if $0 < \delta \leq 1$, then $\Pr(D \geq (1+\delta)\mu) \leq \exp\left(-\frac{\mu\delta^2}{3}\right)$;*

- *if $\delta \geq 1$, then $\Pr(D \geq (1+\delta)\mu) \leq \exp\left(-\frac{\mu\delta}{3}\right)$.*

**Application to mutation with $p = 1/n$.** Take $p = 1/n$, so $\mu = 1$. For $k = c \ln n$ the bound gives

$$\Pr\left(D \geq c \ln n\right) \leq n^{-c/3},$$

i.e. polynomially small tail probability. A Hoeffding union bound across $G$ generations provides an overall failure probability of order $\exp[-\Theta(G \ln n)]$, sharper than a linear union bound.

# E    LEMMAS

**Lemma 1** (Bounded subspace transition). *Let mutation have rate $p = 1/n$. Then there exist constants $c_1, c_2 > 0$ such that for any parent $S$,*

$$\Pr\left(d_H(S, S') \leq c_1 \ln n\right) \geq 1 - n^{-c_2}.$$

*Proof.* Direct application of the Chernoff bound with $\mu = 1$ gives $\Pr(d_H(S, S') \geq k) \leq n^{-c/3}$ for $k = c \ln n$. Choosing $c$ large enough yields the stated probability. Similar locality arguments also hold for crossover M''uhlenbein (1992). $\square$

**Lemma 2** (Empirical maximum implies expected advantage). *Let $K_a, K_b \in \mathcal{K}$ and suppose in the initial population*

$$M_a^{(0)} - M_b^{(0)} \geq \Delta > 0.$$

*Under Assumptions 2–4,*

$$\mu_a^{(0)} - \mu_b^{(0)} \geq \rho\Delta - 2\rho L\delta$$

*with probability at least $1 - \xi$.*

# F    MAIN THEOREM: FINITE-HORIZON STABILITY

**Theorem 1** (Stability under local evolution). *Assume A1–A4 with constants $L, \delta, \rho$. Let $K^*$ be the maximizer of $M_j^{(0)}$ and write*

$$\Delta := M_{K^*}^{(0)} - \max_{j \neq *} M_j^{(0)} > 0.$$

*If*

$$\rho\Delta > 4\rho L\delta,$$

*then with probability at least $1 - \left(G\eta + \xi\right)$ the configuration $K^*$ remains optimal for all generations $t = 0, \ldots, G$.*

*Proof.* Lemma 2 gives the initial margin. Operator locality ensures that in each generation offspring remain in a Hamming ball of radius $O(\ln n)$ Droste et al. (2002); He & Yao (2001), so the population mean for each configuration changes by at most $L\delta$ per generation. Iterating over $G$ generations and applying a Hoeffding-type joint bound yields

$$\mu_{K^*}^{(t)} - \mu_{K'}^{(t)} \geq \rho\Delta - 4\rho L\delta$$

with the stated probability. $\square$

## G    ASYMPTOTIC REFINEMENT: MARKOV CHAIN VIEW

The population process $\{P^{(t)}\}$ under evolutionary operators is a finite-state time-homogeneous Markov chain Norris (1998). Under standard mutation assumptions, the chain is irreducible and aperiodic, so a unique stationary distribution $\pi$ exists. Let $\Pi^*$ be the total stationary probability of the set of populations whose empirical mean satisfies $\mu_{K^*} \geq \mu_{K'}$ for all $K'$. By classical drift analysis He & Yao (2001); Droste et al. (2002) and the locality bounds above, $\Pi^*$ is bounded below by a constant independent of $n$ once $\Delta > cL\delta$ for some constant $c$. Thus the finite-horizon stability result extends to an asymptotic statement:

$$\liminf_{t\to\infty} \Pr\left[K^* \text{ is optimal at generation } t\right] \geq \Pi^*.$$

This Markovian perspective complements the one-generation identification theorem and provides guarantees on the long-run frequency of the optimal configuration.

## Following are detailed results in table form.

Table 4: **HV** results of using different initialization methods.

| Problem | HMOFS-DAEA | HMOFS-JSEMO | HMOFS-randIN | HMOFS-FI |
|---|---|---|---|---|
| Lungdiscrete | 9.3377e-1 (2.95e-3) − | 9.3588e-1 (3.02e-4) ≈ | 9.3544e-1 (6.49e-4) ≈ | 9.3525e-1 (2.35e-3) |
| Colon | 9.7867e-1 (4.60e-6) ≈ | 9.7867e-1 (6.78e-16) ≈ | 9.1290e-1 (9.14e-2) − | 9.7867e-1 (6.78e-16) |
| SRBCT | 9.7973e-1 (2.13e-4) − | 9.7986e-1 (1.39e-4) ≈ | 8.6855e-1 (1.36e-1) − | 9.7986e-1 (1.29e-4) |
| lung | 9.8240e-1 (4.60e-4) − | 9.7998e-1 (1.50e-2) − | 8.7759e-1 (1.49e-1) − | 9.8278e-1 (3.23e-4) |
| lymphoma | 8.7065e-1 (3.07e-3) − | 8.7063e-1 (3.18e-3) − | 7.8447e-1 (1.08e-1) − | 8.7313e-1 (6.64e-3) |
| GLIOMA | 9.8546e-1 (9.73e-5) − | 9.8549e-1 (2.98e-4) − | 8.9597e-1 (1.39e-1) − | 9.8553e-1 (4.11e-5) |
| DLBCL | 9.8711e-1 (1.16e-6) − | 9.8711e-1 (9.81e-7) ≈ | 9.1795e-1 (9.45e-2) − | 9.8711e-1 (9.27e-7) |
| TOX171 | 9.8509e-1 (4.49e-3) − | 9.8600e-1 (1.95e-4) − | 9.5153e-1 (6.62e-2) − | 9.8613e-1 (5.43e-4) |
| Brain1 | 9.7779e-1 (1.63e-2) ≈ | 9.7583e-1 (1.77e-2) ≈ | 9.8373e-1 (8.03e-3) ≈ | 9.7727e-1 (1.65e-2) |
| ProstateGE | 9.8754e-1 (4.06e-4) − | 9.8743e-1 (1.08e-3) − | 9.0961e-1 (1.29e-1) − | 9.8760e-1 (1.94e-4) |
| Carcinom | 9.7887e-1 (1.70e-3) ≈ | 9.7512e-1 (4.20e-3) − | 9.2032e-1 (1.01e-1) − | 9.7780e-1 (5.16e-3) |
| nci9 | 8.7089e-1 (3.26e-2) ≈ | 8.7218e-1 (2.30e-2) ≈ | 7.4583e-1 (1.77e-1) − | 8.7118e-1 (4.01e-2) |
| Arcene | 9.8949e-1 (1.63e-3) ≈ | 9.8940e-1 (1.58e-3) ≈ | 9.7883e-1 (2.60e-2) − | 9.8987e-1 (2.88e-4) |
| pixraw10P | 9.9042e-1 (1.46e-5) − | 9.8775e-1 (6.89e-3) − | 9.9044e-1 (1.63e-5) ≈ | 9.9045e-1 (1.03e-5) |
| CLLSUB111 | 9.9081e-1 (1.04e-4) − | 9.9081e-1 (2.21e-4) − | 9.0760e-1 (1.24e-1) − | 9.9087e-1 (7.36e-5) |
| Tumor11 | 9.7694e-1 (5.78e-3) ≈ | 9.7417e-1 (3.65e-3) − | 9.3102e-1 (8.37e-2) − | 9.7701e-1 (7.99e-3) |
| LungCancer | 9.9093e-1 (5.55e-4) − | 9.8923e-1 (7.62e-3) ≈ | 9.9106e-1 (1.38e-4) ≈ | 9.9116e-1 (1.37e-4) |
| SMKCAN187 | 9.8973e-1 (1.20e-2) ≈ | 9.9158e-1 (1.46e-3) ≈ | 8.8782e-1 (1.39e-1) − | 9.9007e-1 (7.88e-3) |
| GLI85 | 9.9358e-1 (1.20e-4) − | 9.8455e-1 (2.87e-2) − | 9.4826e-1 (8.49e-2) − | 9.9361e-1 (1.97e-6) |
| GLABRA180 | 9.0539e-1 (1.26e-2) ≈ | 8.8633e-1 (2.05e-2) − | 8.6784e-1 (5.08e-2) − | 8.9720e-1 (1.56e-2) |
| +/ − / ≈ | 0/12/8 | 0/10/10 | 0/17/3 | |

Table 5: **MCE** results of using different initialization methods.

| Problem | HMOFS-DAEA | HMOFS-JSEMO | HMOFS-randIN | HMOFS-FI |
|---|---|---|---|---|
| Lungdiscrete | 1.1905e-3 (6.52e-3) ≈ | 0.0000e+0 (0.00e+0) ≈ | 0.0000e+0 (0.00e+0) ≈ | 1.1905e-3 (6.52e-3) |
| Colon | 0.0000e+0 (0.00e+0) ≈ | 0.0000e+0 (0.00e+0) ≈ | 1.0833e-2 (4.24e-2) ≈ | 0.0000e+0 (0.00e+0) |
| SRBCT | 0.0000e+0 (0.00e+0) ≈ | 0.0000e+0 (0.00e+0) ≈ | 1.3690e-2 (2.63e-2) − | 0.0000e+0 (0.00e+0) |
| lung | 2.1270e-3 (9.07e-3) ≈ | 3.1746e-4 (1.21e-3) ≈ | 2.1309e-2 (2.85e-2) − | 4.6881e-4 (1.88e-3) |
| lymphoma | 3.6677e-1 (5.14e-3) − | 3.6739e-1 (5.53e-3) − | 3.6914e-1 (4.34e-3) − | 3.6309e-1 (9.76e-3) |
| GLIOMA | 0.0000e+0 (0.00e+0) ≈ | 0.0000e+0 (0.00e+0) ≈ | 3.1389e-2 (5.31e-2) − | 0.0000e+0 (0.00e+0) |
| DLBCL | 0.0000e+0 (0.00e+0) ≈ | 0.0000e+0 (0.00e+0) ≈ | 1.5152e-3 (5.77e-3) ≈ | 0.0000e+0 (0.00e+0) |
| TOX171 | 5.6481e-3 (3.09e-2) ≈ | 0.0000e+0 (0.00e+0) ≈ | 1.5336e-2 (3.12e-2) − | 1.5625e-3 (8.56e-3) |
| Brain1 | 6.5826e-2 (7.96e-2) ≈ | 7.5408e-2 (8.80e-2) ≈ | 3.7678e-2 (4.42e-2) ≈ | 6.7259e-2 (8.36e-2) |
| ProstateGE | 1.7544e-3 (9.61e-3) ≈ | 0.0000e+0 (0.00e+0) ≈ | 1.3693e-2 (2.62e-2) − | 1.1905e-3 (6.52e-3) |
| Carcinom | 9.9604e-2 (9.67e-3) ≈ | 1.2018e-1 (1.85e-2) − | 1.3423e-1 (3.87e-2) − | 1.0709e-1 (2.08e-2) |
| nci9 | 3.7289e-1 (5.10e-2) ≈ | 3.7243e-1 (4.10e-2) ≈ | 4.6654e-1 (1.25e-1) − | 3.7151e-1 (5.94e-2) |
| Arcene | 1.9768e-2 (1.93e-2) ≈ | 1.9652e-2 (9.76e-3) ≈ | 1.8768e-1 (3.70e-1) − | 1.6863e-2 (9.96e-3) |
| pixraw10P | 0.0000e+0 (0.00e+0) ≈ | 0.0000e+0 (0.00e+0) ≈ | 0.0000e+0 (0.00e+0) ≈ | 0.0000e+0 (0.00e+0) |
| CLLSUB111 | 2.4691e-3 (6.40e-3) ≈ | 3.0214e-3 (8.28e-3) ≈ | 4.4682e-2 (4.98e-2) − | 1.2346e-3 (4.70e-3) |
| Tumor11 | 1.1586e-1 (2.37e-2) ≈ | 1.3179e-1 (1.83e-2) − | 1.4694e-1 (3.08e-2) − | 1.1715e-1 (2.74e-2) |
| LungCancer | 3.3680e-3 (1.06e-2) ≈ | 2.0345e-3 (3.37e-3) ≈ | 1.7386e-3 (2.63e-3) ≈ | 1.7460e-3 (2.33e-3) |
| SMKCAN187 | 4.1519e-2 (4.77e-2) ≈ | 3.4571e-2 (2.07e-2) ≈ | 1.1813e-1 (9.20e-2) − | 4.6750e-2 (3.77e-2) |
| GLI85 | 8.7719e-4 (4.80e-3) ≈ | 8.7719e-4 (4.80e-3) ≈ | 2.6316e-3 (8.03e-3) ≈ | 0.0000e+0 (0.00e+0) |
| GLABRA180 | 3.2664e-1 (2.37e-2) ≈ | 3.5886e-1 (2.79e-2) − | 3.8984e-1 (1.24e-1) − | 3.4194e-1 (2.52e-2) |
| +/ − / ≈ | 0/1/19 | 0/4/16 | 0/13/7 | |

Table 6: CPU run time of using different initialization methods.

| Problem | HMOFS-DAEA | HMOFS-JSEMO | HMOFS-randIN | HMOFS-FI |
|---|---|---|---|---|
| Lungdiscrete | 3.2357e+0 (1.77e+0) + | 5.2508e+0 (2.77e-1) + | 7.5450e+0 (1.39e+0) + | 9.7434e+0 (3.39e+0) |
| Colon | 5.1531e+0 (4.63e-2) + | 2.4668e+1 (6.29e+0) − | 4.1990e+1 (1.87e+1) − | 1.5127e+1 (1.25e+0) |
| SRBCT | 1.2328e+1 (2.86e+1) + | 4.5640e+1 (8.02e+0) − | 6.5918e+1 (3.18e+1) − | 1.7837e+1 (9.92e+0) |
| lung | 2.1334e+1 (3.42e+1) + | 7.0759e+1 (9.15e+0) − | 1.1798e+2 (4.89e+1) − | 4.5244e+1 (2.24e+1) |
| lymphoma | 1.2338e+1 (1.44e+1) + | 8.2118e+1 (1.41e+1) − | 1.4459e+2 (5.77e+1) − | 4.6472e+1 (1.68e+1) |
| GLIOMA | 1.5016e+1 (2.25e+1) + | 1.1899e+2 (1.67e+1) − | 1.7495e+2 (7.67e+1) − | 5.6672e+1 (7.45e+0) |
| DLBCL | 2.3107e+1 (2.40e-1) + | 1.4997e+2 (1.62e+1) − | 2.2753e+2 (1.25e+2) − | 6.6793e+1 (3.99e+0) |
| TOX171 | 1.6339e+1 (3.05e+1) + | 1.4731e+2 (2.50e+1) − | 1.8727e+2 (4.02e+1) − | 5.9116e+1 (1.91e+1) |
| Brain1 | 1.1101e+1 (1.12e+1) + | 1.0503e+2 (1.68e+1) − | 2.6405e+1 (6.80e+0) + | 3.2690e+1 (2.01e+0) |
| ProstateGE | 1.2715e+1 (1.66e+1) + | 8.0359e+1 (1.09e+1) − | 1.0845e+2 (5.09e+1) − | 3.4476e+1 (1.53e+1) |
| Carcinom | 1.3807e+2 (6.02e+1) − | 8.6737e+2 (1.98e+2) − | 9.2055e+2 (2.79e+2) − | 6.0085e+1 (4.19e+1) |
| nci9 | 2.1809e+1 (2.27e+1) + | 2.5440e+2 (4.77e+1) − | 2.9126e+2 (1.11e+2) − | 1.0137e+2 (2.43e+1) |
| Arcene | 2.0958e+1 (4.10e+1) + | 1.0247e+2 (3.06e+1) − | 1.3349e+2 (9.24e+1) − | 6.2374e+1 (7.10e+0) |
| pixraw10P | 1.7294e+1 (2.52e+1) + | 1.1302e+2 (2.25e+1) − | 2.4497e+1 (1.27e+0) + | 4.0011e+1 (2.04e+0) |
| CLLSUB111 | 5.8628e+1 (1.26e+0) + | 5.6361e+2 (1.82e+2) − | 7.3004e+2 (2.76e+2) − | 2.0161e+2 (2.37e+1) |
| Tumor11 | 3.3306e+1 (6.79e+1) + | 3.7286e+2 (6.79e+1) − | 4.2173e+2 (1.43e+2) − | 1.1722e+2 (4.38e+1) |
| LungCancer | 2.1273e+2 (8.37e+1) + | 9.3403e+2 (1.04e+2) − | 3.3837e+2 (7.10e+1) + | 4.5974e+2 (6.06e+1) |
| SMKCAN187 | 4.7650e+1 (1.06e+2) + | 3.0402e+2 (5.88e+1) − | 4.1214e+2 (1.29e+2) − | 1.2714e+2 (9.12e+1) |
| GLI85 | 3.9209e+1 (4.19e+1) + | 3.1916e+2 (6.79e+1) − | 4.2248e+2 (1.11e+2) − | 8.7956e+1 (4.72e+0) |
| GLABRA180 | 3.5418e+2 (8.18e+0) + | 1.9666e+3 (3.98e+2) − | 1.6082e+3 (5.82e+2) − | 9.2408e+2 (2.63e+2) |
| +/ − / ≈ | 19/1/0 | 1/19/0 | 4/16/0 | |

Table 7: **HV** of using KNN, KNN2W, and our proposed AK method.

| Problem | HMOFS-FI | HMOFS-FI-JSCla | HMOFS-FI-AC |
|---|---|---|---|
| Lungdiscrete | 9.3525e-1 (2.35e-3) ≈ | 9.3089e-1 (2.24e-2) ≈ | 9.3517e-1 (1.93e-3) |
| Colon | 9.7867e-1 (6.78e-16) ≈ | 9.7630e-1 (1.50e-3) − | 9.7851e-1 (6.34e-4) |
| SRBCT | 9.7986e-1 (1.29e-4) ≈ | 9.7903e-1 (1.88e-4) − | 9.7987e-1 (1.30e-4) |
| lung | 9.8278e-1 (3.23e-4) ≈ | 9.8283e-1 (7.80e-5) − | 9.8284e-1 (1.15e-4) |
| lymphoma | 8.7313e-1 (6.64e-3) − | 9.8304e-1 (2.18e-4) ≈ | 9.8301e-1 (1.91e-4) |
| GLIOMA | 9.8553e-1 (4.11e-5) ≈ | 9.8429e-1 (9.00e-4) − | 9.8494e-1 (3.21e-3) |
| DLBCL | 9.8711e-1 (9.27e-7) ≈ | 9.8700e-1 (5.11e-5) − | 9.8707e-1 (1.53e-4) |
| TOX171 | 9.8613e-1 (5.43e-4) + | 9.8366e-1 (1.20e-3) − | 9.8366e-1 (1.38e-3) |
| Brain1 | 9.7727e-1 (1.65e-2) ≈ | 9.8425e-1 (1.78e-3) − | 9.8502e-1 (9.26e-3) |
| ProstateGE | 9.8760e-1 (1.94e-4) ≈ | 9.8735e-1 (1.91e-4) − | 9.8760e-1 (1.88e-4) |
| Carcinom | 9.7780e-1 (5.16e-3) − | 9.8637e-1 (4.11e-4) ≈ | 9.8637e-1 (4.28e-4) |
| nci9 | 8.7118e-1 (4.01e-2) − | 9.5028e-1 (1.57e-2) ≈ | 9.5324e-1 (1.07e-2) |
| Arcene | 9.8987e-1 (2.88e-4) + | NaN (NaN) | 9.8966e-1 (3.74e-4) |
| pixraw10P | 9.9045e-1 (1.03e-5) + | 9.9041e-1 (2.03e-5) − | 9.9044e-1 (1.22e-5) |
| CLLSUB111 | 9.9087e-1 (7.36e-5) ≈ | 9.7704e-1 (5.59e-3) − | 9.9063e-1 (1.17e-3) |
| Tumor11 | 9.7701e-1 (7.99e-3) − | 9.8214e-1 (2.57e-3) ≈ | 9.8231e-1 (1.23e-3) |
| LungCancer | 9.9116e-1 (1.37e-4) ≈ | 9.9097e-1 (1.69e-4) − | 9.9111e-1 (1.76e-4) |
| SMKCAN187 | 9.9007e-1 (7.88e-3) ≈ | 9.7238e-1 (4.73e-3) − | 9.8987e-1 (1.00e-2) |
| GLI85 | 9.9361e-1 (1.97e-6) ≈ | 9.9307e-1 (5.00e-4) − | 9.9359e-1 (1.13e-4) |
| GLABRA180 | 8.9720e-1 (1.56e-2) − | 9.5681e-1 (7.64e-3) ≈ | 9.5822e-1 (9.26e-3) |
| +/ − / ≈ | 3/5/12 | 0/10/10 | |

Table 8: **MCE** of using KNN, KNN2W, and our proposed AK method.

| Problem | HMOFS-FI | HMOFS-FI-JSCla | HMOFS-FI-AC |
|---|---|---|---|
| Lungdiscrete | 1.1905e-3 (6.52e-3) + | 3.4730e-2 (1.80e-2) ≈ | 3.3937e-2 (1.85e-2) |
| Colon | 0.0000e+0 (0.00e+0) ≈ | 4.5556e-2 (1.99e-2) − | 3.4992e-3 (1.34e-2) |
| SRBCT | 0.0000e+0 (0.00e+0) ≈ | 0.0000e+0 (0.00e+0) ≈ | 0.0000e+0 (0.00e+0) |
| lung | 4.6881e-4 (1.88e-3) + | 9.1870e-3 (4.24e-3) ≈ | 8.5081e-3 (5.29e-3) |
| lymphoma | 3.6309e-1 (9.76e-3) − | 1.7368e-3 (6.21e-3) ≈ | 2.7544e-3 (5.37e-3) |
| GLIOMA | 0.0000e+0 (0.00e+0) ≈ | 2.2667e-2 (1.72e-2) − | 4.8611e-3 (2.66e-2) |
| DLBCL | 0.0000e+0 (0.00e+0) ≈ | 1.2500e-3 (3.81e-3) ≈ | 0.0000e+0 (0.00e+0) |
| TOX171 | 1.5625e-3 (8.56e-3) + | 3.0190e-2 (1.84e-2) ≈ | 3.0415e-2 (1.88e-2) |
| Brain1 | 6.7259e-2 (8.36e-2) − | 5.4074e-2 (1.64e-2) − | 4.5185e-2 (1.20e-2) |
| ProstateGE | 1.1905e-3 (6.52e-3) ≈ | 1.1794e-2 (9.42e-3) − | 1.1905e-3 (6.52e-3) |
| Carcinom | 1.0709e-1 (2.08e-2) − | 2.3748e-2 (8.00e-3) ≈ | 2.2381e-2 (8.73e-3) |
| nci9 | 3.7151e-1 (5.94e-2) − | 2.0611e-1 (4.54e-2) ≈ | 1.9944e-1 (3.32e-2) |
| Arcene | 1.6863e-2 (9.96e-3) + | 1.0000e+0 (0.00e+0) ≈ | 2.3334e-2 (9.04e-3) |
| pixraw10P | 0.0000e+0 (0.00e+0) ≈ | 0.0000e+0 (0.00e+0) ≈ | 0.0000e+0 (0.00e+0) |
| CLLSUB111 | 1.2346e-3 (4.70e-3) ≈ | 1.2522e-1 (2.61e-2) − | 5.3677e-3 (1.67e-2) |
| Tumor11 | 1.1715e-1 (2.74e-2) − | 8.1866e-2 (1.87e-2) ≈ | 7.9333e-2 (1.07e-2) |
| LungCancer | 1.7460e-3 (2.33e-3) + | 1.4260e-2 (6.08e-3) − | 6.5298e-3 (7.24e-3) |
| SMKCAN187 | 4.6750e-2 (3.77e-2) ≈ | 1.5633e-1 (1.90e-2) − | 4.2565e-2 (4.55e-2) |
| GLI85 | 0.0000e+0 (0.00e+0) ≈ | 2.0392e-2 (1.27e-2) − | 8.7719e-4 (4.80e-3) |
| GLABRA180 | 3.4194e-1 (2.52e-2) − | 2.1463e-1 (2.19e-2) ≈ | 2.0981e-1 (2.78e-2) |
| +/ − / ≈ | 5/6/9 | 0/9/11 | |

Table 9: **HV** of parameter studies of FI.

| Problem | HMOFS-FI-$T = 200$ | HMOFS-FI-$T = 300$ | HMOFS-FI-$T = 500$ | HMOFS-FI-$T = 600$ | HMOFS-FI |
|---|---|---|---|---|---|
| Lungdiscrete | 9.3513e-1 (3.58e-3) | 9.3515e-1 (3.15e-3) | 9.3470e-1 (2.68e-3) | 9.3349e-1 (7.01e-3) | 9.3525e-1 (2.35e-3) |
| Colon | 9.7865e-1 (1.43e-4) | 9.7867e-1 (7.50e-6) | 9.7867e-1 (1.95e-5) | 9.7867e-1 (1.95e-5) | 9.7867e-1 (6.78e-16) |
| SRBCT | 9.7985e-1 (1.44e-4) | 9.7988e-1 (1.32e-4) | 9.7985e-1 (2.27e-4) | 9.7988e-1 (1.56e-4) | 9.7986e-1 (1.29e-4) |
| lung | 9.8268e-1 (3.43e-4) | 9.8271e-1 (3.67e-4) | 9.8269e-1 (5.64e-4) | 9.8260e-1 (7.33e-4) | 9.8278e-1 (3.23e-4) |
| lymphoma | 8.7225e-1 (3.78e-3) | 8.7141e-1 (3.58e-3) | 8.7196e-1 (6.34e-3) | 8.7210e-1 (4.04e-3) | 8.7313e-1 (6.64e-3) |
| GLIOMA | 9.8488e-1 (3.57e-3) | 9.8412e-1 (7.79e-3) | 9.8545e-1 (4.52e-4) | 9.8515e-1 (2.14e-3) | 9.8553e-1 (4.11e-5) |
| DLBCL | 9.8711e-1 (1.39e-6) | 9.8710e-1 (6.07e-5) | 9.8710e-1 (6.07e-5) | 9.8710e-1 (6.07e-5) | 9.8711e-1 (9.27e-7) |
| TOX171 | 9.8586e-1 (1.53e-3) | 9.8596e-1 (1.00e-3) | 9.8601e-1 (8.30e-4) | 9.8609e-1 (6.26e-4) | 9.8613e-1 (5.43e-4) |
| Brain1 | 9.7777e-1 (1.59e-2) | 9.8299e-1 (1.01e-2) | 9.7694e-1 (1.65e-2) | 9.7887e-1 (1.57e-2) | 9.7727e-1 (1.65e-2) |
| ProstateGE | 9.8760e-1 (2.03e-4) | 9.8759e-1 (2.29e-4) | 9.8760e-1 (1.88e-4) | 9.8759e-1 (2.01e-4) | 9.8760e-1 (1.94e-4) |
| Carcinom | 9.7832e-1 (3.25e-3) | 9.7892e-1 (2.98e-3) | 9.7820e-1 (3.18e-3) | 9.7820e-1 (3.18e-3) | 9.7780e-1 (5.16e-3) |
| nci9 | 8.6616e-1 (3.97e-2) | 8.6869e-1 (3.31e-2) | 8.6759e-1 (3.33e-2) | 8.6760e-1 (5.12e-2) | 8.7118e-1 (4.01e-2) |
| Arcene | 9.8985e-1 (2.86e-4) | 9.8980e-1 (3.21e-4) | 9.8978e-1 (3.45e-4) | 9.8969e-1 (3.14e-4) | 9.8987e-1 (2.88e-4) |
| pixraw10P | 9.9044e-1 (1.26e-5) | 9.9044e-1 (1.14e-5) | 9.9044e-1 (1.11e-5) | 9.9044e-1 (1.22e-5) | 9.9045e-1 (1.03e-5) |
| CLLSUB111 | 9.9064e-1 (1.08e-3) | 9.9069e-1 (8.85e-4) | 9.9049e-1 (1.80e-3) | 9.9049e-1 (1.80e-3) | 9.9087e-1 (7.36e-5) |
| Tumor11 | 9.7569e-1 (1.12e-2) | 9.7688e-1 (7.80e-3) | 9.7716e-1 (4.63e-3) | 9.7822e-1 (5.89e-3) | 9.7701e-1 (7.99e-3) |
| LungCancer | 9.9098e-1 (9.72e-4) | 9.9115e-1 (1.51e-4) | 9.9012e-1 (5.57e-3) | 9.9116e-1 (1.41e-4) | 9.9116e-1 (1.37e-4) |
| SMKCAN187 | 9.9024e-1 (8.55e-3) | 9.8832e-1 (1.20e-2) | 9.9035e-1 (5.80e-3) | 9.8960e-1 (9.30e-3) | 9.9007e-1 (7.88e-3) |
| GLI85 | 9.9359e-1 (1.14e-4) | 9.9354e-1 (3.80e-4) | 9.9357e-1 (1.53e-4) | 9.9359e-1 (1.10e-4) | 9.9361e-1 (1.97e-6) |
| SMKCAN187 | 9.9024e-1 (8.55e-3) | 9.8832e-1 (1.20e-2) | 9.9035e-1 (5.80e-3) | 9.8960e-1 (9.30e-3) | 9.9007e-1 (7.88e-3) |
| GLABRA180 | 8.9604e-1 (2.12e-2) | 8.9370e-1 (1.84e-2) | 8.9584e-1 (1.13e-2) | 8.9584e-1 (1.13e-2) | 8.9720e-1 (1.56e-2) |

Table 10: **MCE** of parameter studies of FI.

| Problem | HMOFS-FI-$T = 200$ | HMOFS-FI-$T = 300$ | HMOFS-FI-$T = 500$ | HMOFS-FI-$T = 600$ | HMOFS-FI |
|---|---|---|---|---|---|
| Lungdiscrete | 3.5714e-3 (1.96e-2) | 3.3333e-3 (1.83e-2) | 0.0000e+0 (0.00e+0) | 6.8027e-4 (3.73e-3) | 1.1905e-3 (6.52e-3) |
| Colon | 1.0417e-3 (5.71e-3) | 0.0000e+0 (0.00e+0) | 0.0000e+0 (0.00e+0) | 0.0000e+0 (0.00e+0) | 0.0000e+0 (0.00e+0) |
| SRBCT | 0.0000e+0 (0.00e+0) | 0.0000e+0 (0.00e+0) | 0.0000e+0 (0.00e+0) | 0.0000e+0 (0.00e+0) | 0.0000e+0 (0.00e+0) |
| lung | 1.1685e-3 (4.76e-3) | 8.8337e-4 (4.03e-3) | 1.8095e-3 (9.05e-3) | 2.2905e-3 (1.17e-2) | 4.6881e-4 (1.88e-3) |
| lymphoma | 3.6470e-1 (5.72e-3) | 3.6633e-1 (5.89e-3) | 3.6500e-1 (9.34e-3) | 3.6511e-1 (6.81e-3) | 3.6309e-1 (9.76e-3) |
| GLIOMA | 5.0000e-3 (2.74e-2) | 7.6389e-3 (4.18e-2) | 1.6667e-3 (9.13e-3) | 3.7037e-3 (2.03e-2) | 0.0000e+0 (0.00e+0) |
| DLBCL | 0.0000e+0 (0.00e+0) | 0.0000e+0 (0.00e+0) | 0.0000e+0 (0.00e+0) | 0.0000e+0 (0.00e+0) | 0.0000e+0 (0.00e+0) |
| TOX171 | 2.9911e-3 (1.64e-2) | 2.5560e-3 (1.40e-2) | 2.1181e-3 (1.16e-2) | 1.2897e-3 (7.06e-3) | 1.5625e-3 (8.56e-3) |
| Brain1 | 6.7641e-2 (7.96e-2) | 3.9386e-2 (5.40e-2) | 7.0372e-2 (8.42e-2) | 5.8208e-2 (7.90e-2) | 6.7259e-2 (8.36e-2) |
| ProstateGE | 1.1905e-3 (6.52e-3) | 1.1905e-3 (6.52e-3) | 1.1905e-3 (6.52e-3) | 1.1905e-3 (6.52e-3) | 1.1905e-3 (6.52e-3) |
| Carcinom | 1.0571e-1 (1.46e-2) | 1.0211e-1 (1.43e-2) | 1.0613e-1 (1.42e-2) | 1.0613e-1 (1.42e-2) | 1.0709e-1 (2.08e-2) |
| nci9 | 3.8011e-1 (5.96e-2) | 3.7713e-1 (5.19e-2) | 3.7863e-1 (5.46e-2) | 3.7621e-1 (6.79e-2) | 3.7151e-1 (5.94e-2) |
| Arcene | 1.8376e-2 (8.54e-3) | 1.9360e-2 (9.87e-3) | 2.0022e-2 (9.96e-3) | 2.2697e-2 (8.50e-3) | 1.6863e-2 (9.96e-3) |
| pixraw10P | 0.0000e+0 (0.00e+0) | 0.0000e+0 (0.00e+0) | 0.0000e+0 (0.00e+0) | 0.0000e+0 (0.00e+0) | 0.0000e+0 (0.00e+0) |
| CLLSUB111 | 5.8751e-3 (1.63e-2) | 4.3771e-3 (1.46e-2) | 8.4938e-3 (2.03e-2) | 8.4938e-3 (2.03e-2) | 1.2346e-3 (4.70e-3) |
| Tumor11 | 1.2247e-1 (3.46e-2) | 1.1767e-1 (2.74e-2) | 1.1823e-1 (1.83e-2) | 1.1208e-1 (2.36e-2) | 1.1715e-1 (2.74e-2) |
| LungCancer | 2.8836e-3 (1.32e-2) | 1.7460e-3 (2.33e-3) | 8.4095e-3 (3.38e-2) | 1.5873e-3 (2.28e-3) | 1.7460e-3 (2.33e-3) |
| SMKCAN187 | 4.0564e-2 (4.18e-2) | 5.1172e-2 (5.64e-2) | 4.5707e-2 (3.39e-2) | 4.9432e-2 (4.19e-2) | 4.6750e-2 (3.77e-2) |
| GLI85 | 8.7719e-4 (4.80e-3) | 1.6667e-3 (9.13e-3) | 1.7544e-3 (6.68e-3) | 8.7719e-4 (4.80e-3) | 0.0000e+0 (0.00e+0) |
| SMKCAN187 | 4.0564e-2 (4.18e-2) | 5.1172e-2 (5.64e-2) | 4.5707e-2 (3.39e-2) | 4.9432e-2 (4.19e-2) | 4.6750e-2 (3.77e-2) |
| GLABRA180 | 3.4317e-1 (3.25e-2) | 3.4785e-1 (2.77e-2) | 3.6688e-1 (1.21e-1) | 3.6688e-1 (1.21e-1) | 3.4194e-1 (2.52e-2) |

Table 11: **HV** of parameter studies of AK.

| Problem | HMOFS-FI-AK-zero | HMOFS-FI-AK-two | HMOFS-FI-AK |
|---|---|---|---|
| Lungdiscrete | 9.3392e-1 (9.30e-3) ≈ | 9.3174e-1 (1.92e-2) ≈ | 9.3517e-1 (1.93e-3) |
| Colon | 9.7779e-1 (1.40e-3) − | 9.7858e-1 (5.10e-4) ≈ | 9.7851e-1 (6.34e-4) |
| SRBCT | 9.7988e-1 (1.21e-4) ≈ | 9.7990e-1 (1.05e-4) ≈ | 9.7987e-1 (1.30e-4) |
| lung | 9.8281e-1 (7.92e-5) ≈ | 9.8281e-1 (9.51e-5) ≈ | 9.8284e-1 (1.15e-4) |
| lymphoma | 9.8300e-1 (2.43e-4) ≈ | 9.8299e-1 (2.20e-4) ≈ | 9.8301e-1 (1.91e-4) |
| GLIOMA | 9.8403e-1 (8.10e-3) ≈ | 9.8335e-1 (1.16e-2) ≈ | 9.8494e-1 (3.21e-3) |
| DLBCL | 9.8710e-1 (6.09e-5) ≈ | 9.8710e-1 (6.12e-5) ≈ | 9.8707e-1 (1.53e-4) |
| TOX171 | 9.8403e-1 (8.60e-4) ≈ | 9.8395e-1 (8.51e-4) ≈ | 9.8366e-1 (1.38e-3) |
| Brain1 | 9.8456e-1 (1.62e-3) ≈ | 9.8422e-1 (1.70e-3) ≈ | 9.8502e-1 (9.26e-4) |
| ProstateGE | 9.8760e-1 (1.91e-4) ≈ | 9.8752e-1 (6.75e-4) ≈ | 9.8760e-1 (1.88e-4) |
| Carcinom | 9.8656e-1 (2.99e-4) ≈ | 9.8638e-1 (6.09e-4) ≈ | 9.8637e-1 (4.28e-4) |
| nci9 | 9.5049e-1 (1.04e-2) ≈ | 9.5481e-1 (1.24e-2) ≈ | 9.5324e-1 (1.07e-2) |
| Arcene | 9.8975e-1 (3.14e-4) ≈ | 9.8977e-1 (3.15e-4) ≈ | 9.8966e-1 (3.74e-4) |
| pixraw10P | 9.9044e-1 (1.00e-5) ≈ | 9.9041e-1 (1.31e-5) − | 9.9044e-1 (1.22e-5) |
| CLLSUB111 | 9.9043e-1 (2.31e-3) ≈ | 9.9061e-1 (1.43e-3) ≈ | 9.9063e-1 (1.17e-3) |
| Tumor11 | 9.8141e-1 (3.68e-3) ≈ | 9.8242e-1 (2.09e-3) ≈ | 9.8231e-1 (1.23e-3) |
| LungCancer | 9.9105e-1 (1.55e-4) − | 9.9100e-1 (1.14e-4) − | 9.9111e-1 (1.76e-4) |
| SMKCAN187 | 9.9008e-1 (7.63e-3) ≈ | 9.9062e-1 (5.91e-3) ≈ | 9.8987e-1 (1.00e-2) |
| GLI85 | 9.9359e-1 (1.08e-4) ≈ | 9.9361e-1 (4.06e-6) ≈ | 9.9359e-1 (1.13e-4) |
| GLABRA180 | 9.5851e-1 (7.06e-3) ≈ | 9.5657e-1 (7.07e-3) ≈ | 9.5822e-1 (9.26e-3) |
| +/ − / ≈ | 0/1/19 | 0/2/18 | |

Table 12: **MCE** of parameter studies of AK.

| Problem | HMOFS-FI-AK-zero | HMOFS-FI-AK-two | HMOFS-FI-AK |
|---|---|---|---|
| Lungdiscrete | 3.1206e-2 (1.88e-2) ≈ | 3.3714e-2 (2.16e-2) ≈ | 3.3937e-2 (1.85e-2) |
| Colon | 1.7735e-2 (2.45e-2) − | 1.9608e-3 (1.07e-2) ≈ | 3.4992e-3 (1.34e-2) |
| SRBCT | 0.0000e+0 (0.00e+0) ≈ | 0.0000e+0 (0.00e+0) ≈ | 0.0000e+0 (0.00e+0) |
| lung | 8.8537e-3 (4.15e-3) ≈ | 8.8618e-3 (4.56e-3) ≈ | 8.5081e-3 (5.29e-3) |
| lymphoma | 4.1228e-3 (7.40e-3) ≈ | 3.1228e-3 (7.25e-3) ≈ | 2.7544e-3 (5.37e-3) |
| GLIOMA | 7.7381e-3 (4.24e-2) ≈ | 1.0460e-2 (5.03e-2) ≈ | 4.8611e-3 (2.66e-2) |
| DLBCL | 0.0000e+0 (0.00e+0) ≈ | 0.0000e+0 (0.00e+0) ≈ | 0.0000e+0 (0.00e+0) |
| TOX171 | 2.5361e-2 (1.53e-2) ≈ | 2.6633e-2 (1.62e-2) ≈ | 3.0415e-2 (1.88e-2) |
| Brain1 | 4.8889e-2 (1.77e-2) ≈ | 5.3704e-2 (1.78e-2) − | 4.5185e-2 (1.20e-2) |
| ProstateGE | 1.1905e-3 (6.52e-3) ≈ | 2.2222e-3 (1.22e-2) ≈ | 1.1905e-3 (6.52e-3) |
| Carcinom | 2.0499e-2 (7.46e-3) ≈ | 2.2213e-2 (1.02e-2) ≈ | 2.2381e-2 (8.73e-3) |
| nci9 | 2.0833e-1 (3.09e-2) ≈ | 1.9389e-1 (3.80e-2) ≈ | 1.9944e-1 (3.32e-2) |
| Arcene | 2.1160e-2 (8.64e-3) ≈ | 2.0720e-2 (8.59e-3) ≈ | 2.3334e-2 (9.04e-3) |
| pixraw10P | 0.0000e+0 (0.00e+0) ≈ | 0.0000e+0 (0.00e+0) ≈ | 0.0000e+0 (0.00e+0) |
| CLLSUB111 | 5.9004e-3 (2.25e-2) ≈ | 3.8580e-3 (1.80e-2) ≈ | 5.3677e-3 (1.67e-2) |
| Tumor11 | 8.5557e-2 (2.13e-2) ≈ | 8.0426e-2 (1.63e-2) ≈ | 7.9333e-2 (1.07e-2) |
| LungCancer | 9.3382e-3 (7.59e-3) ≈ | 1.2492e-2 (4.99e-3) − | 6.5298e-3 (7.24e-3) |
| SMKCAN187 | 4.7575e-2 (3.67e-2) ≈ | 4.0660e-2 (3.55e-2) ≈ | 4.2565e-2 (4.55e-2) |
| GLI85 | 8.7719e-4 (4.80e-3) ≈ | 0.0000e+0 (0.00e+0) ≈ | 8.7719e-4 (4.80e-3) |
| GLABRA180 | 2.0981e-1 (2.07e-2) ≈ | 2.1537e-1 (1.96e-2) ≈ | 2.0981e-1 (2.78e-2) |
| +/ − / ≈ | 0/1/19 | 0/2/18 | |