# OpenReview forum: "Fast and Adaptive Multi-Objective Feature Selection for Classification"
_ICLR.cc/2026/Conference — ICLR 2026 Conference Withdrawn Submission_

### Official Review · Reviewer_fbYB · 2025-10-21

**Soundness:** 2
**Presentation:** 2
**Contribution:** 2
**Rating:** 2
**Confidence:** 4

**Summary:**

The paper states that to address bottlenecks of evolutionary wrapper based multi objective feature selection, including time inefficiency, an exponentially growing search space, and the need to adaptively choose a classifier, the authors propose two simple yet effective methods: FI and AK. FI uses mutual information with tournament selection to rapidly identify high quality initial feature subsets. AK provides a theoretical guarantee that a single generation is enough to adaptively determine the best KNN for different datasets, with minimal time overhead and without extra data analysis or assumptions. Experiments on 20 real world high dimensional datasets show that FI and AK outperform advanced initialization and KNN schemes, and the selected feature subsets transfer well to a tabular LLM, enabling seamless application to high dimensional data with superior results.

**Strengths:**

Compared with other studies that use evolutionary computation for feature selection, this paper adds theoretical validation on top of the experimental work. Although it only demonstrates “the most search compatible KNN configuration for a given dataset after only one generation of evolutionary evaluation, when the feature space is high dimensional and the search budget is limited,” this still improves the paper’s readability.

**Weaknesses:**

1.The number of external baseline algorithms is very limited; the experiments are not sufficiently comprehensive, and the external comparisons are weak.
2.The authors claim the proposed method is fast, but it is slower than DAEA, and the authors’ explanation asserting superiority over HMOFS-DAEA is not persuasive. Based on this, I do not consider the method “fast.”
3.Across the experiments on eight datasets, AK sometimes even yields negative gains (performance degradation).
4. Adaptive KNN is an existing method in the literature.

**Questions:**

1.On what grounds do the authors regard the proposed algorithm as Fast? What is their reasoning for this claim?
2.The paper states that “TabPFN can finish its inference in a few seconds on all these datasets.” What is the evidence for this statement? The paper does not seem to report runtime measurements.
3.There are many feature selection approaches beyond EC-based wrapper methods, including filter and embedded methods. Why were these not included as baselines, especially given the small number of external comparisons in the current experiments?
4.The introduction notes that adaptive KNN has already been proposed. Why does the paper not compare against existing adaptive KNN methods?

---

### Official Review · Reviewer_5cHM · 2025-10-26

**Soundness:** 2
**Presentation:** 2
**Contribution:** 2
**Rating:** 2
**Confidence:** 4

**Summary:**

This paper presents a novel framework for **Fast and Adaptive Multi-Objective Feature Selection (MOFS)** for high-dimensional data classification, aiming to overcome challenges such as time inefficiency, large search space, and the difficulty of adaptively selecting a suitable classifier in wrapper-based evolutionary MOFS methods.

The core contribution consists of two simple yet effective methods integrated into the HMOFS-FI-AK framework:

1.  **Fast Initialization (FI):** This method calculates **Mutual Information (MI)** between each feature and the class label to gauge feature importance. It utilizes a **parameter-adaptive tournament selection** mechanism to efficiently locate high-quality initial feature subsets, ensuring a balance between feature importance and population diversity while maintaining high computational efficiency.
2.  **one-generation Adaptive K-Nearest Neighbor (AK):** AK addresses the challenge of fixing the K-Nearest Neighbor (KNN) settings in complex data. Through theoretical proof, the authors verify that the most suitable KNN configuration (from a set of candidates) can be determined after evaluating only a **single evolutionary generation** of independent populations, thereby improving feature selection performance with minimal time overhead and without requiring prior data analysis or assumptions.

The MOFS problem is formalized as a bi-objective optimization: minimizing the **balanced classification error** ($f_1(x)$) to avoid biases towards majority classes, and minimizing the **selected feature ratio** ($f_2(x)$). Experiments on 20 high-dimensional real-world datasets demonstrate that FI and AK achieve superior performance compared to existing advanced methods. Crucially, the feature subsets selected by this method generalize well to the LLM TabPFN, enabling its seamless and rapid application to high-dimensional data (which it otherwise cannot handle due to a strict limit of $\le 500$ features) while achieving superior performance.

**Strengths:**

*   **One-Generation Adaptive Classifier (AK):** Introducing an adaptive classifier (KNN) within wrapper-based MOFS is typically infeasible due to the computational cost required at every iteration. AK's core originality lies in its **one-generation determination** of the optimal KNN, which provides coherence between feature quality and the classifier. This strategy is validated by **theoretical proof** (Theorem 1), establishing the stability of the selection across multiple generations given initial conditions.

*   **Bridging MOFS and LLMs:** A significant outcome is the generalization capability of the resulting feature subsets. The subsets obtained by HMOFS-FI-AK enable the specialized LLM TabPFN (which strictly requires feature numbers $\le 500$) to be seamlessly applied to high-dimensional datasets, achieving superior accuracy and rapid inference time. This is highly important for extending the utility of powerful models to complex, large-scale data environments.

**Weaknesses:**

1.  **Limited KNN Configuration Space for AK:**
    AK's effectiveness is demonstrated by selecting between only two fixed KNN configurations: **KNN ($k=5$)** and **KNN2W (JSCla)**. While the adaptability is proven within this scope, real-world high-dimensional data exhibit diverse characteristics (e.g., class imbalance, complex distributions). The restriction to only two configurations, particularly $k=5$ and a single fixed weighted KNN variant (KNN2W), may not sufficiently validate AK's ability to find the *globally* optimal classifier setting across highly distinct datasets.
    *   *Actionable Insight:* Expanding the candidate set $K = \{K_1, \dots, K_m\}$ to include KNN configurations with smaller $k$ values (e.g., $k=1$ or $k=3$) or different distance metrics would strengthen the claim of "automatically determining the most suitable KNN".

2.  **Fixed Sparse Factor ($T$) Lacks Dataset Adaptivity:**
    The sparse factor $T$ in FI, which controls the sparsity of the initial population, is set to a fixed value of **$T=400$** for all 20 experiments. The sources show these datasets vary drastically in dimensionality, ranging from 325 to 49,151 features. Although sensitivity analysis indicates $T=400$ performs best overall, a fixed value may not be ideally tailored for datasets at the extremes of this dimensional range, potentially limiting the initial population quality for specific problems.
    *   *Actionable Insight:* The authors should explore and suggest a heuristic or mechanism for adaptively setting $T$ based on a quantifiable property of the dataset, such as its total dimensionality ($d$) or calculated metrics, rather than maintaining a universal fixed value.

3.  **Cost of Adaptation vs. Evolutionary Budget:**
    The paper notes that the slightly inferior performance results observed for HMOFS-FI-AK (compared to the best fixed KNN on certain specific datasets) are due to the fact that the multiple independent populations used in the first generation of AK consume a **part of the very limited evaluation budget** ($E_{max} = 20000$ evaluations total). This highlights that even the minor overhead introduced by AK can slightly compromise the overall search trajectory given the fixed, strict evaluation budget typical in MOFS problems.
    *   *Actionable Insight:* While the gain from adaptation usually outweighs this loss, the authors could further analyze if there are strategies (e.g., dynamic budget allocation or smarter initial sampling in AK) that could mitigate this marginal loss of search budget in the initial phase.

**Questions:**

1.  **Applicability of AK to Heterogeneous Classifiers:**
    The core of AK relies on the theoretical proof (Theorem 1) derived from assumptions like Lipschitz sensitivity and operator locality, which support the stability of the optimal KNN choice during limited evolution. If the selection set $K = \{K_1, \dots, K_m\}$ were extended to include **heterogeneous classifiers** (e.g., choosing between KNN, a Support Vector Machine, or a Decision Tree), would the underlying theoretical assumptions of AK regarding *local Lipschitz sensitivity* and *finite-horizon stability* still hold true, or would the differing nature of classification functions require a fundamental revision of the theoretical guarantee?

2.  **Practical Interpretation of Theoretical Margins ($\rho, \Delta, L, \delta$):**
    Theorem 1 confirms stability if the initial margin satisfies **$\rho\Delta > 4\rho L\delta$**. This condition involves parameters related to the population coverage ($\rho$), Hamming ball radius ($\delta$), Lipschitz constant ($L$), and empirical accuracy gap ($\Delta$). Could the authors provide practical guidance on how these theoretical parameters, which are crucial for the stability guarantee, can be estimated or verified empirically in real-world high-dimensional MOFS problems, particularly since distance-based classification performance deteriorates in high dimensions?

3.  **Discretization Sensitivity in FI:**
    FI's success hinges on accurately calculating Mutual Information (MI). For continuous features, the paper states they are discretized using **equal-frequency with 10 bins**. Was this choice of 10 bins subjected to a sensitivity analysis? Since the number of bins significantly influences the empirical estimation of MI, could a poor choice potentially undermine the selection of truly high-quality features in the initial population?

4.  **Optimal Pareto Front Solution for TabPFN Transfer:**
    The final feature subsets are used to test performance on TabPFN. Since the MOFS process produces a Pareto Front (PF) of trade-off solutions, which specific criterion (e.g., minimum classification error $f_1$, minimum size $f_2$, or a point like the knee of the PF) was used to select the "final feature subset" for transfer to TabPFN in Table 2, given that decision-makers prioritize different objectives? Understanding this criterion is important for generalizing the TabPFN application approach.

---

### Official Review · Reviewer_jmBG · 2025-10-29

**Soundness:** 2
**Presentation:** 2
**Contribution:** 1
**Rating:** 2
**Confidence:** 5

**Summary:**

The paper proposes two techniques, Fast Initialization (FI) and Adaptive KNN (AK), to improve multi-objective feature selection for high-dimensional classification. The authors argue that FI quickly seeds the evolutionary population with important features, while AK determines the best KNN configuration using only one generation of evolution. Experiments on several real-world datasets are presented, including claimed improvements when applying selected features to a tabular foundation model.

**Strengths:**

The motivation—high dimensionality, time cost, and adaptability issues in KNN classification—is well articulated.

FI is conceptually straightforward and computationally lightweight, avoiding O(d²N) correlation computations.

Experimental evaluation spans 20 datasets and includes statistical significance testing.

An attempt is made to apply selected feature subsets to TabPFN to demonstrate the benefits of generalization.

**Weaknesses:**

1) FI is essentially MI ranking + randomized tournament selection—neither new nor theoretically insightful for MOFS. AK is a single-generation model selection heuristic wrapped inside EA iterations; despite lengthy “theoretical justifications,” it lacks convincing novelty or rigor.

2) The AK argument relies on strong assumptions such as Lipschitz properties of classifier accuracy and proximity to local optima in the initial population—these are unrealistic for high-dimensional discrete search. The proof sketch resembles a generic EA drift analysis inserted into the context, with no dataset-specific or algorithm-dependent guarantees.

3) 69/30 train-test split performed only once; no cross-validation is used, making HV and MCE comparisons unreliable, especially on datasets with <200 samples.

4) The adaptive KNN uses training error to pick a classifier, which may bias toward overfitting and distort generalization metrics.

5) Only a subset of competing initialization methods is evaluated on high-dimensional data due to time constraints, leading to unfair comparisons. The TabPFN results do not support the narrative. In multiple datasets (Carcinom, SMK—CAN—187), FI-AK is not the best-performing method, contradicting strong claims in the abstract.

6) Gains seem dataset-dependent and potentially due to chance rather than methodological superiority.

7) There are grammatical errors, redundant explanations, and inconsistent use of technical terminology. The flowchart (Fig. 2) contains confusing arrows and unclear execution order. Evaluation metrics do not fully support MOFS. Balanced error is used, but class-imbalance complexity is not analyzed, despite being part of the motivation.

8) Pareto-front comparisons rely solely on hypervolume; no diversity or robustness measures are included.

**Questions:**

How is the theoretical claim of “one-generation identifiability” justified in real-world settings without the restrictive assumptions stated?

Why is a single 70/30 split used? How stable are results under k-fold cross-validation or repeated splits?

How was the parameter T = 400 chosen for FI? Is the performance sensitive to dataset dimensionality?

Why not include more recent/stronger baselines for initialization and classifier adaptation, particularly those that scale with high dimensions?

Does AK truly reduce time overhead? Running multiple parallel populations in generation 1 seems more expensive than using a single pre-selected classifier.

TabPFN shows multiple cases where FI or FI-JSCla outperform FI-AK—how does that support the paper’s main contribution?

What happens when the best KNN configuration changes during evolution—a scenario the method explicitly ignores?

---

### Official Review · Reviewer_Bnsi · 2025-10-31

**Soundness:** 3
**Presentation:** 3
**Contribution:** 2
**Rating:** 4
**Confidence:** 4

**Summary:**

The paper proposes a framework for multi-objective feature selection (MOFS) in high-dimensional classification tasks. It addresses the computational burden and classifier-selection limitations of evolutionary wrapper methods by introducing: (1) Fast Initialization, which uses mutual information and adaptive tournament sampling to efficiently generate diverse, high-quality initial feature subsets; (2) Adaptive KNN, which determines the most suitable KNN variant in only one evolutionary generation with theoretical justification. Integrated into an evolutionary MOFS algorithm, these methods deliver improved Pareto fronts, reduced classification error, and significantly lower runtime across 20 real-world high-dimensional datasets. The selected features also transfer effectively to a tabular LLM, TabPFN, enabling good performance on datasets too large for direct model input, demonstrating both efficiency and practical applicability.

**Strengths:**

(1)The paper has well motivations, and proposes simple but effective methods, FI and AK.

(2)The proposed FI significantly reduces initialization cost; AK selects KNN variant with negligible overhead.

(3)The paper did comprehensive empirical evaluations, 20 high-dimensional real-world datasets are used, with multiple baselines and statistical tests.

**Weaknesses:**

(1)The novelty of the proposed methods is incremental, as both the mutual-information-based initialization and adaptive KNN selection extend existing ideas rather than introducing fundamentally new algorithmic paradigms.

(2)The adaptive classifier mechanism is limited to KNN variants, leaving unclear whether the approach generalizes to broader classifier families commonly used in feature-selection pipelines (e.g., SVMs, tree-based models).

(3)The theoretical justification for one-generation KNN selection relies on assumptions that may not always hold in real-world high-dimensional search spaces, making the analysis partly heuristic.

(4)The empirical validation is heavily focused on biological high-dimensional datasets and a single MOEA framework, and broader evaluations across diverse domains and additional evolutionary baselines would strengthen the generality claims.

**Questions:**

(1)How sensitive is the proposed method to the choice of mutual information estimation and discretization strategy, especially for continuous or noisy high-dimensional features?

(2)Can the adaptive classifier selection framework be extended beyond KNN (e.g., to SVM or tree-based classifiers), and if so, how would the one-generation selection strategy scale?

---

### Note · Authors · 2025-11-28

I have read and agree with the venue's withdrawal policy on behalf of myself and my co-authors.